# δ-Lactones—A New Class of Compounds That Are Toxic to *E. coli* K12 and R2–R4 Strains

**DOI:** 10.3390/ma14112956

**Published:** 2021-05-30

**Authors:** Paweł Kowalczyk, Barbara Gawdzik, Damian Trzepizur, Mateusz Szymczak, Grzegorz Skiba, Stanisława Raj, Karol Kramkowski, Rafał Lizut, Ryszard Ostaszewski

**Affiliations:** 1Department of Animal Nutrition, The Kielanowski Institute of Animal Physiology and Nutrition, Polish Academy of Sciences, 05-110 Jabłonna, Poland; g.skiba@ifzz.pl (G.S.); s.raj@ifzz.pl (S.R.); 2Institute of Chemistry, Jan Kochanowski University, Swietokrzyska 15 G, 25-406 Kielce, Poland; 3Institute of Organic Chemistry PAS, Kasprzaka 44/52, 01-224 Warsaw, Poland; damian.trzepizur@icho.edu.pl (D.T.); ryszard.ostaszewski@icho.edu.pl (R.O.); 4Department of Molecular Virology, Institute of Microbiology, Faculty of Biology, University of Warsaw, Miecznikowa 1, 02-096 Warsaw, Poland; mszymczak@biol.uw.edu.pl; 5Department of Physical Chemistry, Medical University of Bialystok, Kilińskiego 1 str, 15-089 Bialystok, Poland; kkramk@wp.pl; 6Department of Mathematics, Informatics and Landscape Architecture, Institute of Informatics, The John Paul II Catholic University of Lublin, 20-708 Lublin, Poland; lizut@kul.pl

**Keywords:** δ-lactones, *E. coli* strains, oxidative stress, antibiotics, lipopolysaccharide

## Abstract

Lactones are among the well-known organic substances with a specific taste and smell. They are characterized by antibacterial, antiviral, anti-inflammatory, and anti-cancer properties. In recent years, among this group of compounds, new biologically active substances have been searched by modifying the main (leading) structure with new analogs with stronger or different responses that may have a toxic effect on the cells of pathogenic bacteria and constitute an alternative to commonly used antibiotics. A preliminary study of δ-lactone derivatives as new potential candidates for antibacterial drugs was conducted. Particular emphasis was placed on the selection of the structure of lactones with the highest biological activity, especially those with fluorine in their structure as a substituent in terms of action on bacterial lipopolysaccharide (LPS) in the model strains of *Escherichia coli* K12 (without LPS in its structure) and R2–R4 (LPS of different lengths in its structure). In the presented studies, on the basis of the conducted MIC and MBC tests, it was shown that the antibacterial (toxic) activity of lactones depends on their structure and the length of the bacterial LPS in the membrane of specific strains. Moreover, oxidative damage of bacterial DNA isolated from bacteria after modification with newly synthesized compounds after application of the repair enzyme Fpg glycosylase was analyzed. The analyzed damage values were compared with the modification with appropriate antibiotics: ciprofloxacin, bleomycin, and cloxacillin. The presented research clearly shows that lactone derivatives can be potential candidates as substitutes for drugs, e.g., the analyzed antibiotics. Their chemical and biological activity is related to coumarin derivatives and the corresponding δ-lactone groups in the structure of the substituent. The observed results are of particular importance in the case of increasing bacterial resistance to various drugs and antibiotics, especially in nosocomial infections and neoplasms, and in the era of a microbial pandemic.

## 1. Introduction

Lactones are a group of cyclic esters formed as a result of intramolecular esterification of hydroxy acids (Figure 1). General structure of lactones **A** (depicted in Figure 1) shows the most important structural features of these molecules. It is important to note the presence of endocyclic ester group in target structure (Figure 1) [1]. Depending on the mutual position of the hydroxyl and carboxyl groups in the starting hydroxy acid, basic four types of lactones are distinguished: α, β, γ (structure B), δ (structure **D**), and ε (structure **C**), differing in the size of the cyclic ring (Figure 1).

Particular types of lactones show different durability and properties. The most stable are γ-lactones and δ-lactones, i.e., with five or six cyclic ring atoms, respectively. The lowest stability can be attributed to α-lactones, with three cyclic ring atoms, due to the strong torsion and angular stresses in the bonds between the atoms forming the ring. Similar stresses, to a slightly lesser extent, are also observed in β-lactones, with four cyclic ring atoms. Compounds with a lactone group containing more than six ring atoms are usually unstable due to the unfavorable interactions associated with the structure of the cyclic system [1,2,3,4].

One of the typical lactones is coumarin and its derivatives, which have a broad antibacterial effect, causing damage to the bacterial cell membrane, e.g., in *E. coli* strains [5,6,7,8,9,10,11]. The K12 strain of *E. coli* belongs to the model organisms among bacteria because its structure, genetics, and metabolism are well known and used in genetic studies. It is a Gram-negative relatively anaerobic bacterium belonging to the Enterobacteriaceae family. It is part of the physiological bacterial flora of the human colon and other warm-blooded animals. In the intestine, this symbiotic bacterium plays a useful role by participating in the breakdown of food and also contributing to the production of B and K vitamins. Under certain conditions, the colon rod is pathogenic for humans, mainly causing diseases of the digestive and urinary systems. It does not have LPS, unlike the analyzed mutants of *E. coli* R1–R4, which have a complete LPS core region composed of the outer and inner core [5,6,7,8,9,10,11]. The most important properties of coumarin and its derivatives include anti-cancer, antiviral, anti-allergic, anti-inflammatory, and anti-thrombotic properties [12]. Coumarins are also used as inhibitors of plant, bacterial, and fungal growth [13,14,15]. Important representative of ε-lactones is ε-Caprolactone C. This colorless liquid compound, miscible with water and several organic solvents, is produced in large quantities in chemical industry as an important polymer caprolactam precursor. A good representative of seven-membered ring δ-lactone is 6-phenyl tetrahydro pyran-2-one of structure D (Figure 1), [1].

The knowledge on antimicrobial properties of lactones is still fragmentary, therefore, there is a need to clarify their role. From the lithographic data, it is known only that bacteriostatic properties are exhibited by substances in which the lactone moiety is present in a small ring, e.g., xanthatin [16], a bicyclic lactone isolated from *Xanthium pensylvanicum* and *Xanthium strumarium*, which is active against *Staphylococcus aureus*, including MRSA-resistant methicillin strains [16,17]. Antimicrobial properties are also demonstrated by α-methylene-γ-butyrolactones condensed with a 10-membered ring. The most representative examples are: tavulin, dentin A, tanachin, isolated from *Tanacetum argyrophyllum, or Nocardia*, showing activity against *Staphylococcus aureus*, *Escherichia coli*, and *Bacillus subtilis* ATCC 6633 [18,19,20,21,22,23,24]. Antitumor properties are demonstrated by macrocyclic lactones and numerous bicyclic compounds with a multi-membered ring fused with the α-methylene-γ-butyrolactone ring. Macrocyclic lactones are used as drugs in the treatment of cancers of the breast, ovary, testes, and pancreas. They induce tubulin aggregation and formation of multipolar spindles and consequently, stop mitosis in the G2 phase [23,24]. Antibiotics with a cytostatic effect containing a lactone group are actinomycins with an α-methylene-γ-butyrolactone ring in their structure produced by *Actinomycetes antibioticus* cultures (also known as streptomycins and *Streptomyces antibioticus*).

Actinomycins work by modifying the cell cycle that slows down cell progression through the S phase and stops cell division in the G2 phase. Such properties are shown, e.g., by actinomycin D (3) [25,26]. This antibiotic is composed of two cyclopentapeptides in which the lactone bond connects the threonine hydroxyl group with the valine carboxyl group. Substituted four-membered ring lactones often exhibit enzyme-inhibitory properties. For example, we can mention lipstatin [27] isolated from *Streptomyces toxytricini* and its saturated analogue tetrahydrolipstatin [28]. These both compounds are pancreatic lipase inhibitors, which makes them useful in the treatment of obesity. The described activity is related to the presence in the structure of these compounds of a reactive β-lactone ring, which acylates the serine hydroxyl group in the catalytic center of the lipase [29]. Antiviral properties are demonstrated by condensed polycyclic systems containing an α-methylene-γ-butyrolactone ring. These include centaurepensin (chlorohysopifoline A) [30], chloroanerine [31], and 13-acetylsolstitialin, isolated from the aerial parts of *Centaurea solstitialis* [32]. They are mainly used in the treatment of viral infections of the upper respiratory tract, diseases of the digestive system, and herpes caused by HPV 2 (genital herpes virus) [33]. Artemisinin (20), a sesquiterpene lactone found in *Artemisia annua* [25,26,27,28,29,30,31,32,33] exhibits antimalarial properties. Fe^2+^ ions from hemoglobin exert an influence on the activity of this lactone. Some lactones have anti-inflammatory properties. These include the sesquiterpene lactone helenalin [34], extracted from *Arnica Montana* and *Arnica chamissonis ssp. Foliosa*. For many years, an alcoholic extract of arnica containing helenalin and its derivatives has been used in traditional medicine to treat hematomas, rheumatic diseases, and superficial skin inflammations. Helenalin is an inhibitor of the NF-κB transcription factor, which plays a key role in regulating the immune response to infections. This lactone affects the NF-κB proteins present in T cells and causes the activation of the mitochondrial apoptotic pathway and the arrest of the cell cycle in the G2/M phase, which, in turn, contributes to a significant reduction in the immune response [35,36] (Figure 1).

The aim of the study is to analyze the potential antibacterial activity of lactones as potential antibiotics in terms of their structure, especially those containing fluorine as a substituent, in terms of cytotoxic effect on bacterial lipopolysaccharides (LPS) in model strains used in this type of research of *Escherichia coli* K12 (without LPS in its structure) and R2–R4 (LPS of different lengths in its structure).

## 2. Materials and Methods

### 2.1. Microorganisms and Media

Bacterial strains were obtained as a kind gift from Prof. Jolanta Łukasiewicz (Polish Academy of Sciences, Wrocław, Poland). All compounds used in the experiments were the same as described in [5,6,7,8,9].

### 2.2. Experimental Chemistry

All other reagents used in this work were purchased from (Sigma-Aldrich Saint Louis, Missouri, USA). ^1^H NMR and ^13^C NMR spectra were recorded in solutions (CDCl_3_ with c.a. 1% TMS) on Bruker Avance DRX 600 spectrometer. IR spectra were recorded on a PerkinElmer Spectrum 400. HR-ESI-MS spectra were recorded on Bruker micrOTOF-Q II with ionization ESI. Analytical gas chromatography (GC) was performed on a Thermo Scientific-Trace 1310 apparatus using TG-5HT (30 m × 0.25 mm) capillary column. Analytical thin layer chromatography (TLC) was carried out on silica gel G (Merck), and various developing systems were applied. Compounds detected by I_2_ in the iodide chamber. Column chromatography was performed on silica gel (Kiselgel 60, 230–400 mesh, Merck) with hexane-ethyl acetate and hexane-acetone. Chemical shifts are expressed in parts per million using TMS as an internal standard.

### 2.3. General Procedure for the Synthesis of Lactones

Procedure for synthesis of unsaturated secondary alcohols 3

The synthesis of unsaturated secondary alcohols **3** (Figure 2) was carried out following the procedure described by others and Gaylord and Becker [36,37,38,39,40,41,42,43,44,45,46,47,48,49,50,51,52,53,54,55,56,57,58,59,60,61,62]. A solution of aryl bromide **1** (0.08 mol) in diethyl ether (Figure 2) was added dropwise to magnesium turnings (0.08 mol) and heated under reflux for 4 h. The mixture was cooled on ice and a solution of 3-methyl crotonaldehyde **2** in diethyl ether (Figure 2) was added dropwise and stirred for 24 h. The mixture was dissolved in diethyl ether and added to ice and water. The separated organic layer was dried over anhydrous magnesium sulfate and evaporated under vacuum. The crude products were purified by column chromatography on silica gel (ethyl acetate/hexane, 1:8 (*v*/*v*)) (Figure 2).

Procedure for synthesis of γ,δ-unsaturated carboxylic acids 4

A mixture of alcohol **3** (0.01 mol), ethyl orthoacetate (20 cm^3^, 0.11 mol), and propionic acid (0.1 cm^3^, 0.001 mol) was heated (138 °C) for 5 h with simultaneous distilling of the ethanol formed. Then, orthoacetate was distilled off, and the crude product was purified by column chromatography (hexane-acetone, 3:1). After purification, compounds were directly used for subsequent reaction. KOH (0.61 g) solution in ethanol (25 mL) was added to ester (0.0076 mol). The reaction mixture was refluxed for 2 h. The course of the reaction was monitored by TLC. Then, ethanol was distilled off and the crude product was dissolved in water (20 mL) and washed with diethyl ether (100 mL). Next, the aqueous layer was added to 0.1 M HCl, and the crude product was extracted with ether (7 × 50 mL). The ethereal solution was dried over MgSO_4_, and the solvent was evaporated to give pure acids 4 (Figure 2).

Procedure for synthesis of iodolactones 5 and 6

Briefly, 0.5 M NaHCO_3_ (25 mL) was added to a solution of carboxylic acid 4a–c (0.0070 mol) in diethyl ether (25 mL). The reaction mixture was stirred at room temperature for 30 min. Then, iodine (3.6 g, 0.0284 mol) and KI (7.0 g, 0.0422 mol) in water (30 mL) was added dropwise to a refluxed reaction mixture. The reaction mixture was refluxed and stirred for 5 h. Then, the mixture was cooled to room temperature, diluted with ethyl ether (50 mL), and washed with an aqueous saturated solution of sodium thiosulfate. The ethereal solution was washed with an aqueous saturated solution of NaHCO_3_ and brine, and dried over MgSO_4_. The crude products were purified by column chromatography (hexane-ethyl acetate, 5:1) (Figure 2).

Procedure for synthesis of chlorolactones 5 and 6

The mixture of an unsaturated carboxylic acid 4a–c (0.005 mol) and *N*-chlorosuccinimide (0.009 mol) was dissolved in 60 mL of THF. Acetic acid was added dropwise, and the reaction mixture was stirred at room temperature for 48 h. When the reaction was completed, the mixture was dissolved in diethyl ether, washed with saturated NaHCO_3_, and dried over anhydrous magnesium sulfate. The crude product was purified by column chromatography on silica gel using a mixture of acetone and hexane (1:10) (Figure 2).

Procedure for synthesis of bromolactones 5 and 6

The mixture of an unsaturated carboxylic acid 4a–c (0.005 mol) and N-bromosuccinimide (0.01 mol) was dissolved in 60 mL of THF. Acetic acid was added dropwise and reaction mixture was stirred at room temperature for 48 h. When the reaction was completed, the mixture was dissolved in diethyl ether, washed with saturated NaHCO_3_, and dried over anhydrous magnesium sulfate. The crude product was purified by column chromatography on silica gel using a mixture of acetone and hexane (1:10).

#### 2.3.1. Product 5a 5-Chloro-4-Methyl-6-Phenyltetrahydro-2H-Pyran-2-One

Colorless crystals m.p. 105–106.5 °C, R_f_ = 0.18 (acetone:hexan 1:7); ^1^H NMR(CDCl_3_, 500 MHz) δ (ppm): 1.25 (d, 3H, J = 6.4 Hz, CH(CH_3_)), 2.35–2.47 (m, 2H, J = 17.2 Hz, CHaH_b_), 2.92–3.01(m, 1H, CH_a_H_b_), 3.76 (t, 1H, J = 9.8Hz, CHCl), 5.18 (d, 1H, J = 9.8 Hz, CHAr), 7.33–7.43 (m, 5H, H_Ar_); ^13^C NMR (125 MHz, CDCl_3_) δ (ppm): 19.6 (CH(CH_3_)), 35.2 (CH(CH_3_)_2_), 37.1 (CH_2_), 63.3 (CHCl), 85.0 (CHAr), 168.9; (C=O), C_Ar_: 127.51, 128.5, 129.2, 136.4; IR (cm^−1^): 1730, 1496, 1460, 1401, 1376, 1345, 1288, 1240, 1218, 1184, 1107, 1074, 1055, 1031, 1020, 995, 930, 906, 893, 850, 800, 757, 670, 651, 618;. ESI-TOF HR: *m*/*z* calcd for C_12_H_13_ClO_2_: 247.0502 [M+Na]^+^, found 247.0510.

#### 2.3.2. Product 5b 5-Bromo-4-Methyl-6-Phenyltetrahydro-2H-Pyran-2-One

Colorless crystals mp. 111–112 °C, R_f_ = 0.16 (acetone:hexan 1:7); ^1^H NMR (CDCl_3_, 500 MHz) δ (ppm): 1.28 (d, 3H, J = 6.5 Hz, CH(CH_3_)), 2.44 (dd, 1H, J_1_ = 17.2 Hz, J_2_ = 10.0 Hz, CH_a_H_b_), 2.47–2.59 (m, 1H, CH(CH_3_)), 2.99 (dd, 1H, J_1_ = 17.2 Hz, J_2_ = 5.6 Hz, CH_a_H_b_), 3.86 (dd, 1H, J_1_ = 10.2 Hz, J_2_ = 9.7 Hz, CHBr), 5.32 (d, 1H, J = 10.3 Hz, CHAr), 7.34–7.44 (m, 5H, H_Ar_); ^13^C NMR (125 MHz, CDCl_3_) δ (ppm): 21.0 (CH(CH_3_)), 35.8 (CH(CH_3_)_2_), 37.4 (CH_2_), 55.4 (CHBr), 85.4 (CHAr), 169.2 (C=O), C_Ar_: 127.6, 128.5, 129.6, 136.7;. ESI-TOF HR: *m*/*z* calcd for C_12_H_13_BrO_2_: 290.9996 [M+Na]^+^, found 291.0007.

#### 2.3.3. Product 5c 5-Chloro-4,4-Dimethyl-6-Phenyltetrahydro-2H-Pyran-2-One

Colorless crystals mp. 96–97 °C, R_f_ = 0.24 (ethyl acetate:hexan 1:7); ^1^H NMR (CDCl_3_, 500 MHz) δ (ppm): 1.19 (s, 3H, C(CH_3_)_2_), 1.26 (s, 3H, C(CH_3_)_2_), 2.57 (d, 1H, J = 17.4 Hz, CH_a_H_b_), 2.76 (d, 1H, J = 17.4 Hz, CH_a_H_b_), 3.97 (d, 1H, J = 10.4 Hz, CHCl), 5.25 (d, 1H, J = 10.4 Hz, CH_Ar_); ^13^C NMR (125 MHz, CDCl_3_) δ (ppm): 20.4 (C(CH_3_)_2_), 28.2 (C(CH_3_)_2_), 35.48 (C(CH_3_)_2_), 44.2 (CH_2_), 67.2 (CHCl), 83.0 (CHAr), 168.3 (C=O), C_Ar_: 127.4, 128.5, 129.2, 136.9; ESI-TOF HR: *m*/*z* calcd for C_13_H_15_ClO_2_: 261.0658 [M+Na]^+^, found 261.0651.

#### 2.3.4. Product 5d 5-Bromo-4,4-Dimethyl-6-Phenyltetrahydro-2H-Pyran-2-One

Colorless crystals mp. 107–108 °C, R_f_ = 0.25 (ethyl acetate:hexan 1:7); ^1^H NMR (CDCl_3_, 500 MHz) δ (ppm): 1.20 (s, 3H, C(CH_3_)_2_), 1.28 (s, 3H, C(CH_3_)_2_), 2.61 (d, 1H, J = 17.4 Hz, CH_a_H_b_), 2.81 (d, 1H, J = 17.4 Hz, CH_a_H_b_), 4.11 (d, 1H, J = 10.8 Hz, CHBr), 5.40 (d, 1H, J = 10.8 Hz, CHAr), 7.35–7.42 (m, 5H, H_Ar_); ^13^C NMR (125 MHz, CDCl_3_) δ (ppm): 21.4 (C(CH_3_)), 29.6 (C(CH_3_)_2_), 35.5 (C(CH_3_)_2_), 44.0 (CH_2_), 61.3 (CHBr), 83.3 (CHAr), 168.4 (C=O), C_Ar_: 127.5, 128.5, 129.2, 137.1; ESI-TOF HR: *m*/*z* calcd for C_13_H_15_BrO_2_: 305.0153 [M+Na]^+^, found 305.0138.

#### 2.3.5. Product 5e 5-Chloro-4-Methyl-6-(4-Fluorophenyl)-Tetrahydro-2H-Pyran-2-One

Colorless crystals mp. 80–81 °C, R_f_ = 0.16 (acetone:hexan 1:7); ^1^H NMR (CDCl_3_, 500 MHz) δ (ppm): 1.25 (d, 3H, J = 6.3 Hz, CH(CH_3_)), 2.34–2.49 (m, 2H, CH_a_H_b_, CH(CH_3_)), 2.94–3.05 (m, 1H, CH_a_H_b_), 3.71 (t, 1H, J = 9.8 Hz, CHCl), 5.17 (d, 1H, J = 10.0 Hz, CHAr), 7.03–7.13 (m, 2H, H_Ar_), 7.32–7.42 (m, 2H, H_Ar_); ^13^C NMR (125 MHz, CDCl_3_) δ (ppm): 19.6 (CH(CH_3_)), 35.5 (CH(CH_3_)), 37.0 (CH_2_), 63.2 (CHCl), 84.3 (CHAr), 168.7 (C=O), C_Ar_: 115.5 (d, ^2^J_F-C_ = 21.9 Hz), 129.3 (d, ^3^J_F-C_ = 8.4 Hz), 132.3 (d, ^4^J_F-C_ = 3.2 Hz), 163.1 (d, J_F-C_ = 248.4 Hz); ESI-TOF HR: *m*/*z* calcd for C_12_H_12_ClFO_2_: 265.0407 [M+Na]^+^, found 265.0410.

#### 2.3.6. Product 5f. 5-Bromo-4-Methyl-6-(4-Fluorophenyl)-Tetrahydro-2H-Pyran-2-One

Colorless crystals mp. 123–124 °C, R_f_ = 0.12 (acetone:hexan 1:7); ^1^H NMR (CDCl_3_, 500 MHz) δ (ppm): 1.28 (d, 3H, J = 6.5 Hz, CH(CH_3_)), 2.44 (dd, 1H, J_1_ = 17.2 Hz, J_2_ = 10.0 Hz, CH_a_H_b_), 2.47–2.61 (m, 1H, CH(CH_3_)), 2.99 (dd, 1H, J_1_ = 17.2 Hz, J_2_ = 5.6 Hz, CH_a_H_b_), 3.80 (dd, 1H, J_1_ = 10.4 Hz, J_2_ = 9.7 Hz, CHBr), 5.31 (d, 1H, J = 10.4 Hz, CHAr), 7.04–7.16 (m, 2H, H_Ar_), 7.32–7.42 (m, 2H, H_Ar_); ^13^C NMR (125 MHz, CDCl_3_) δ (ppm): 21.0 (CH(CH_3_)), 35.8 (CH(CH_3_)), 37.3 (CH_2_), 55.5 (CHBr), 84.6 (CHAr), 168.9 (C=O), C_Ar_: 115.5 (d, ^2^J_F-C_ = 21.8 Hz), 129.4 (d, ^3^J_F-C_ = 8.5 Hz), 132.7 (d, ^4^J_F-C_ = 3.3 Hz), 163.1 (d, J_F-C_ = 248.6 Hz); ESI-TOF HR: *m*/*z* calcd for C_12_H_12_BrFO_2_: 308.9902 [M+Na]^+^, found 308.9916.

#### 2.3.7. Product 5g. 5-Iodo-4-Methyl-6-(4-Fluorophenyl)-Tetrahydro-2H-Pyran-2-One

Colorless crystals mp. 67–68 °C, R_f_ = 0.16 (acetone:hexan 1:7); ^1^H NMR (CDCl_3_, 500 MHz) δ (ppm): 1.28 (d, 3H, J = 6.5 Hz, CH(CH_3_)), 2.36 (dd, 1H, J_1_ = 17.2 Hz, J_2_ = 10.0 Hz, CH_a_H_b_), 2.51–2.57 (m, 1H, CH(CH_3_)), 2.92 (dd, 1H, J_1_ = 17.5 Hz, J_2_ = 6.0 Hz, CH_a_H_b_), 3.90 (dd, 1H, J_1_ = 10.8 Hz, J_2_ = 10.1 Hz, CHI), 5.42 (d, 1H, J = 10.8 Hz, CHAr), 7.05–7.09 (m, 2H, H_Ar_), 7.32–7.36 (m, 2H, H_Ar_); ^13^C NMR (125 MHz, CDCl_3_) δ (ppm): 23.6 (CH(CH_3_)), 27.03 (CH(CH_3_)_2_), 31.66 (CH_2_), 37.69 (CHI), 86.43 (CHAr), 169.68 (C=O), C_Ar_: 115.53 (d, ^2^J_F-C_ = 21.7 Hz), 129.58 (d, ^3^*J_F-C_* = 8.4 Hz), 133.78 (d, ^4^J*_F-C_*= 3.3 Hz), 163.09 (d, J_F-C_ = 248.6 Hz); ESI-TOF HR: *m*/*z* calcd for C_12_H_12_FIO_2_: 356.9763 [M+Na]^+^, found 356.9777.

#### 2.3.8. Product 5h. 5-chloro-4,4-Dimethyl-6-(4-Fluorophenyl)-Tetrahydro-2H-Pyran-2-One

Colorless crystals mp. 89–90 °C, R_f_ = 0.22 (ethyl acetate:hexan 1:7); ^1^H NMR (CDCl_3_, 500 MHz) δ (ppm): 1.19 (s, 3H, C(CH_3_)_2_), 1.26 (s, 3H, C(CH_3_)_2_), 2.56 (d, 1H, J = 17.4 Hz, CH_a_H_b_), 2.76 (d, 1H, J = 17.4 Hz, CH_a_H_b_), 3.92 (d, 1H, J = 10.5 Hz, CHCl), 5.24 (d, 1H, J = 10.5 Hz, CHAr), 6.98–7.14 (m, 2H, H_Ar_), 7.31–7.42 (m, 2H, H_Ar_); ^13^C NMR (125 MHz, CDCl_3_) δ (ppm): 20.4 (C(CH_3_)_2_), 28.2 (C(CH_3_)_2_), 35.38 (C(CH_3_)_2_), 44.2 (CH_2_), 67.2 (CHCl), 82.2 (CHAr), 168.1 (C=O), C_Ar_: 115.5 (d, ^2^*J*_F-C_ = 21.8 Hz), 129.3 (d, ^3^J_F-C_ = 8.5 Hz), 132.8 (d, ^4^J_F-C_ = 3.6 Hz), 163.0 (d, J_F-C_ = 248.4 Hz); ESI-TOF HR: *m*/*z* calcd for C_13_H_14_ClFO_2_: 279.0564 [M+Na]^+^, found 279.0575.

#### 2.3.9. Product 5i 5-Bromo-4,4-Dimethyl-6-(4-Fluorophenyl)-Tetrahydro-2H-Pyran-2-One

Colorless crystals mp. 102–103 °C; R_f_ = 0.23 (ethyl acetate:hexan 1:7); ^1^H NMR (CDCl_3_, 500 MHz) δ (ppm): 1.21 (s, 3H, C(CH_3_)_2_), 1.28 (s, 3H, C(CH_3_)_2_), 2.60 (d, 1H, J = 17.4 Hz, CH_a_H_b_ 2.82 (d, 1H, J = 17.4 Hz, CH_a_H_b_), 4.05 (d, 1H, J = 10.8 Hz, CHBr), 5.39 (d, 1H, J = 10.8 Hz, CHAr), 7.03–7.12 (m, 2H, H_Ar_), 7.32–7.39 (m, 2H, H_Ar_); ^13^C NMR (125 MHz, CDCl_3_) δ (ppm): 21.3 (C(CH_3_)_2_), 29.5 (C(CH_3_)_2_), 35.3 (C(CH_3_)_2_), 44.0 (CH_2_), 61.3 (CHBr), 82.6 (CHAr), 168.2 (C=O), C_Ar_: 115.5 (d, ^2^J_F-C_= 21.9 Hz), 129.4 (d, ^3^J_F-C_= 8.6 Hz), 133.1 (d, ^4^J_F-C_= 3.4 Hz), 163.0 (d, J_F-C_ = 248.6 Hz); ESI-TOF HR: *m*/*z* calcd for C_13_H_14_BrFO_2_: 323.0068 [M+Na]^+^, found 323.0080.

#### 2.3.10. Product 5j. 5-Iodo-4,4-Dimethyl-6-(4-Fluorophenyl)-Tetrahydro-2H-Pyran-2-One

Yellow crystals mp. 81–82 °C, R_f_ = 0.18 (ethyl acetate:hexan 1:7); ^1^H NMR (CDCl_3_, 500 MHz) δ (ppm): 1.20 (s, 3H, C(CH_3_)_2_), 1.28 (s, 3H, C(CH_3_)_2_), 2.64 (d, 1H, J = 17.3 Hz, CH_a_H_b_), 2.76 (d, 1H, J = 17.3 Hz, CH_a_H_b_), 4.22 (d, 1H, J = 11.4 Hz, CHI), 5.54 (d, 1H, J = 11.4 Hz, CHAr), 7.04–7.14 (m, 2H, H_Ar_), 7.32–7.35 (m, 2H, H_Ar_); ^13^C NMR (125 MHz, CDCl_3_) δ (ppm): 23.2 (C(CH_3_)_2_), 32.5 (C(CH_3_)_2_), 35.4 (C(CH_3_)_2_), 42.3 (CHI), 45.0 (CH_2_), 84.1 (CHAr), 168.6 (C=O), C_Ar_: 115.5 (d, ^2^J_F-C_ = 21.8 Hz), 129.5 (d, ^3^J_F-C_ = 8.5 Hz), 133.7 (d, ^4^J_F-C_ = 3.3 Hz), 163.0 (d, J_F-C_ = 248.7 Hz); ESI-TOF HR: *m*/*z* calcd for C_13_H_14_FIO_2_: 370.9920 [M+Na]^+^, found 371.0010.

#### 2.3.11. Product 6a 5-Chloro-4-Methyl-6-Naphtyltetrahydro-2H-Pyran-2-One

Colorless crystals mp. 157–158 °C, R_f_ = 0.19 (acetone:hexan 1:7); ^1^H NMR (CDCl_3_, 500 MHz) δ (ppm): 1.29 (d, 3H, J = 6.4 Hz, CH(CH_3_)), 2.47–2.61 (m, 2H, CH(CH_3_), CH_a_H_b_), 3.07 (dd, 1H, J_1_ = 18.5 Hz, J_2_ = 10.0 Hz, CH_a_H_b_), 4.18 (t, 1H, J = 9.7 Hz, CHCl), 5.99 (d, 1H, J = 10.0 Hz, CHAr), 7.47–7.61 (m, 4H, H_Ar_), 7.90 (d, 2H, J = 8.3 Hz, H_Ar_), 8.09 (d, 1H, J = 8.5 Hz, H_Ar_); ^13^C NMR (125 MHz, CDCl_3_) δ (ppm): 19.7 (CH(CH_3_)), 35.9 (CH(CH_3_)), 37.1 (CH_2_), 62.7 (CHCl), 81.9 (CHAr), 169.0 (C=O), C_Ar_: 123.0, 125.0, 125.8, 125.9, 126.6, 129.0, 130.1, 131.1, 131.8, 133.8; ESI-TOF HR: *m*/*z* calcd for C_16_H_15_ClO_2_: 313.0397 [M+Na]^+^, found 371.0010.

#### 2.3.12. Product 6b. 5-Bromo-4-Methyl-6-Naphtyltetrahydro-2H-Pyran-2-One

Colorless crystals mp. 168–169 °C, R_f_ = 0.13 (acetone:hexan 1:7); ^1^H NMR (CDCl_3_, 500 MHz) δ (ppm): 1.33 (d, 3H, J = 6.6 Hz, CH(CH_3_)), 2.55 (dd, 1H, J_1_ = 17.4 Hz, J_2_ = 10.1 Hz, CH_a_H_b_), 2.59–2.72 (m, 1H, CH(CH_3_)), 3.08 (dd, 1H, J_1_ = 17.4 Hz, J_2_ = 5.7 Hz, CH_a_H_b_), 4.28 (dd, 1H, J_1_ = 10.3 Hz, J_2_ = 9.6 Hz, CHBr), 6.12 (d, 1H, J = 10.3 Hz, CHAr), 7.49–7.53 (m, 2H, H_Ar_), 7.53–7.59 (m, 4H, H_Ar_), 7.88–7.93 (m, 2H, H_Ar_), 8.10 (d, 1H, J = 8.6 Hz, H_Ar_); ^13^C NMR (125 MHz, CDCl_3_) δ (ppm): 21.1 (CH(CH_3_)), 36.3 (CH(CH_3_)), 37.4 (CH_2_), 54.7 (CHBr), 82.1 (CHAr), 169.2 (C=O), C_Ar_: 123.0, 125.0, 125.8, 126.0, 126.6, 129.1, 130.1, 131.1, 132.2, 133.9; ESI-TOF HR: *m*/*z* calcd for C_16_H_15_BrO_2_: 341.0153 [M+Na]^+^, found 341.0169.

#### 2.3.13. Product 6c 5-Iodo-4-Methyl-6-Naphtyltetrahydro-2H-Pyran-2-One

Yellow crystals mp. t.t. = 126–127 °C, R_f_ = 0.16 (acetone:hexan 1:7); 1H NMR (CDCl_3_, 500 MHz) δ (ppm): 1.31 (d, 3H, J = 6.4 Hz, CH(CH_3_)), 2.53–2.67 (m, 2H, CH(CH_3_), CH_a_H_b_), 3.14 (dd, 1H, J_1_ = 18.2 Hz, J_2_ = 10.0 Hz, CH_a_H_b_),4.23 (t, 1H, J = 9.7 Hz, CHI), 6.09 (d, 1H, J = 10.0 Hz, CHAr), 7.44–7.51 (m, 4H, H_Ar_), 7.60–7.66 (m, 2H, H_Ar_), 8.09 (d, 1H, J = 8.6 Hz, H_Ar_); ^13^C NMR (125 MHz, CDCl_3_) δ (ppm): 23.1 (CH(CH_3_)), 29.8 (CH(CH_3_)), 32.5 (CH_2_), 38.6 (CHI), 84.3 (CHAr), 169.2 (C=O), C_Ar_: 118.0, 125.1, 125.8, 125.9, 127.0, 129.1, 130.2, 131.2, 132.1, 133.9; ESI-TOF HR: *m*/*z* calcd for C_16_H_15_IO_2_: 366.1943 [M+Na]^+^, found 366.1945.

#### 2.3.14. Product 6d. 5-Chloro-4,4-Dimethyl-6-Naphtyltetrahydro-2H-Pyran-2-One

Colorless crystals mp. 192–193 °C, R_f_ = 0.20 (ethyl acetate:hexan 1:7); 1H NMR (CDCl_3_, 500 MHz) δ (ppm): 1.23 (s, 3H, C(CH_3_)_2_), 1.35 (s, 3H, C(CH_3_)_2_), 2.68 (d, 1H, J = 17.4 Hz, CH_a_H_b_), 2.84 (d, 1H, J = 17.4 Hz, CH_a_H_b_), 4.37 (d, 1H, J = 10.4 Hz, CHCl), 6.06 (d, 1H, J = 10.4 Hz, CHAr), 7.47–7.60 (m, 4H, H_Ar_), 7.88 (d, 2H, J = 7.8 Hz, H_Ar_), 8.05 (d, 1H, J = 8.5 Hz, H_Ar_); ^13^C NMR (125 MHz, CDCl_3_) δ (ppm): 20.8 (C(CH_3_)_2_), 29.5 (C(CH_3_)_2_), 35.7 (C(CH_3_)_2_), 44.3 (CH_2_), 66.6 (CHCl), 79.5 (CHAr), 168.5 (C=O), C_Ar_: 122.9, 125.0, 125.8, 125.9, 126.6, 129.1, 130.0, 131.2, 132.2, 133.8; ESI-TOF HR: *m*/*z* calcd for C_17_H_17_ClO_2_: 311.0814 [M+Na]^+^, found 311.0806.

#### 2.3.15. Product 6e. 5-Bromo-4,4-Dimethyl-6-Naphtyltetrahydro-2H-Pyran-2-One

Colorless crystals mp. 197–198 °C, R_f_ = 0.20 (ethyl acetate:hexan 1:7); 1H NMR (CDCl_3_, 500 MHz) δ (ppm): 1.29 (s, 3H, C(CH_3_)_2_), 1.42 (s, 3H, C(CH_3_)_2_), 2.76 (d, 1H, J = 17.4 Hz, CH_a_H_b_), 2.94 (d, 1H, J = 17.4 Hz, CH_a_H_b_), 4.56 (d, 1H, J = 10.7 Hz, CHBr), 6.25 (d, 1H, J = 10.7 Hz, CHAr), 7.47–7.53 (m, 2H, H_Ar_), 7.53–7.59 (m, 2H, H_Ar_), 7.89 (d, 2H, J = 8.2 Hz, H_Ar_), 8.08 (d, 1H, J = 8.6 Hz, H_Ar_); ^13^C NMR (125 MHz, CDCl_3_) δ (ppm): 21.7 (C(CH_3_)_2_), 29.6 (C(CH_3_)_2_), 35.6 (C(CH_3_)_2_), 44.1 (CH_2_), 60.5 (CHBr), 79.6 (CHAr), 168.5 (C=O), C_Ar_: 122.9, 125.0, 125.8, 125.9, 126.6, 129.1, 130.1, 131.2, 132.5, 133.8; ESI-TOF HR: *m*/*z* calcd for C_17_H_17_BrO_2_: 371.0048 [M+Na]^+^, found 371.0040.

#### 2.3.16. Product 6f. 5-Iodo-4,4-Dimethyl-6-Naphtyltetrahydro-2H-Pyran-2-One

Yellow crystals mp. 118–119 °C, R_f_ = 0.16 (ethyl acetate:hexan 1:7); 1H NMR (CDCl_3_, 500 MHz) δ (ppm): 1.26 (s, 3H, C(CH_3_)_2_), 1.40 (s, 3H, C(CH_3_)_2_), 2.78 (d, 1H, J = 17.3 Hz, CH_a_H_b_), 2.96 (d, 1H, J = 17.3 Hz, CH_a_H_b_), 4.71 (d, 1H, J = 11.4 Hz, CHI), 6.36 (d, 1H, J = 11.4 Hz, CHAr), 7.48–7.55 (m, 2H, H_Ar_), 7.56–7.61 (m, 2H, H_Ar_), 7.90 (d, 2H, J = 8.2 Hz, H_Ar_), 8.10 (d, 1H, J = 8.6 Hz, H_Ar_); ^13^C NMR (125 MHz, CDCl_3_) δ (ppm): 23.6 (C(CH_3_)_2_), 32.5 (C(CH_3_)_2_), 35.7 (C(CH_3_)_2_), 42.5 (CHI), 43.5 (CH_2_), 77.2 (CHAr), 168.9 (C=O), C_Ar_: 122.9, 124.9, 125.9, 125.9, 126.6, 129.1, 130.2, 131.2, 133.1, 133.9; ESI-TOF HR: *m*/*z* calcd for C_17_H_17_IO_2_: 403.0170 [M+Na]^+^, found 403.0172.

### 2.4. Cytotoxicity Study of Bacterial Cells—Application of MIC and MBC Tests

The analysis of the minimum inhibitory concentration (MIC) and minimum bactericidal concentration (MBC) was performed on the basis of the original method described in detail others and by Prost [36,37,38,39,40,41,42,43,44,45,46,47,48,49,50,51,52,53,54,55,56,57,58,59,60,61,62,63] with its individual modifications for specific compounds analyzed and described in the works [5,6,7,8,9].

### 2.5. Isolation Plasmids DNA from Bacterial K12 and R2–R4 Strains

All four bacterial DNA were isolated according to New England Biolabs, Lab-JOT protocols as described in [5,6,7,8,9].

#### 2.5.1. Interaction of the Plasmid DNA from K12 and R4 Strains with Peptones

Interactions of the Plasmid DNA from K12 and R4 strains with the analyzed lactones were described in details in the same way as that of coumarin derivatives, α-amidoamides, and 1,2-diarylethanols [5,6,7].

#### 2.5.2. Interaction of the Plasmid DNA from K12 and R4 Strains with Selected Antibiotics

The analysis of antibiotics such as cloxacillin was carried out in the same way as that of α-amidoamides as described earlier in the literature [6].

### 2.6. Cleavage of Plasmid DNA by Application of Fpg Glycosylases in Bacterial Cells

The cleavage of plasmid DNA by application of Fpg glycosylases was carried out in the same way as described earlier in the literature [5,6,7,8,9].

### 2.7. Cleavage of Plasmid DNA by Fpg Protein Modified by Selected Antibiotics

The modified DNA bacterial plasmids modified by selected antibiotic were digested by Fpg enzyme to show similar interaction as that of the analyzed diaryloalcohols. The method of cleavage of plasmid DNA by Fpg protein is described in detail in literature [6].

### 2.8. Statistical Analysis

Statistical analyses were conducted using Statistica software (version 12, StatSoft, Tulsa, OK, USA). The parametric analyses using Tukey’s test was used to determine differences between groups at *p* < 0.05, * *p* < 0.01 **, and *p* < 0.001 ***. The number of experimental repetitions presented in each figure was a minimum of three.

## 3. Results

### 3.1. Chemistry

The biological activity of six-membered lactones (δ-lactones) is far less investigated with respect to lactams. It is probably due to the fact that synthesis of lactams is much more expanded [36,37,38,39,40]. Several methods used for lactams synthesis, such as one-pot, BF_3_·OEt_2_-mediated reaction of various substituted arenes with azido alkanoic acid chlorides [41], irydium catalyzed aminolysis of lactones and amines via aminolysis of lactone, N-alkylation of amine with hydroxyamide, intramolecular transamidation of aminoamide transformation [42], or a metal-free selective oxidation of cyclic secondary and tertiary amines using molecular iodine [42], cannot be used for lactones synthesis.

Commonly used procedures for δ-lactones synthesis were recently discussed in literature in details [43]. Lactones are an important group of small-molecule signaling compounds found in different organisms such as plants, insects, vertebrates, and microorganisms [43]. Although the antimicrobial activity of lactones was already mentioned in literature [44], the γ-halo-δ-lactones were far less investigated [45]. This is presumable due to the fact that synthesis of halo lactones is much more complicated and cumbersome. In order to highlight the role of halogen atom on the biological activity of δ-lactones, appropriate unsaturated carboxylic acids of structure **4** are required. These compounds are hardly available as chemical building blocks and must be prepared in multistep syntheses. Several synthetic procedures consist Wittig-Horner [46], cross metathesis reactions [47], or Johnson-Claisen rearrangement of allyl esters [48]. Recent studies showed that δ-decalactones Ia inhibited the growth of *Aspergillus niger, Candida albicans*, and *Staphylococcus aureus* pathogens [49] and might be useful as protective agents against fungi and viruses in the air. This compound possessed also interesting in vitro antifungal activity against *Penicillium digitatum*, *P. expansum P. italicum*, *Botrytis cinerea,* and *Sclerotinia* sp. [50]. The synthesis of target lactones **5** and **6** was performed according to the procedure depicted in Figure 2.

The reaction between Grignard reagents generated in situ from aryl bromides **1** and magnesium in ester solution, and crotonaldehydes **2**, led to formation of vinyl alcohols **3**. In the next reaction, triethyl orthoacetate was added into the solution of alcohols **3** in propionic acid and reaction mixture was heated at 138 °C to complete the Johnson–Claisen rearrangement to the respective ethyl esters, which were directly hydrolyzed using potassium hydroxide solution to give carboxylic acids **4** in an overall yield of 60%. The last halo lactonization of acids 4  with N-bromosuccinimide, N-chlorosuccinimide, or iodide in potassium iodide solution gives final halogeno-lactones **5** and **6** in 37–93% yield. The individual lactones **5** and **6** (Figure 3) were isolated via column chromatography. It is interesting to note that single **δ**-lactones were obtained in 37–93% yield (Table 1).

The structure of all compounds was confirmed using NMR and mass spectroscopy, and for some compounds also the correct elemental analysis data were recorded (Table 1).

### 3.2. Toxicity of Tested Compounds

For the antibacterial analysis of δ-lactones, as in the case of studies on coumarin derivatives, alpha-amidoamides, 1,2-diaryl alcohols, and ionic liquids, MIC and MBC tests are used [5,6,7,8,9]. Both types of MIC and MBC tests are based on the study of the permeability and damage to the bacterial cell membrane by the analyzed compounds, which may lead to their apoptosis. It is visible by changing the color of the analyzed sample with a positively charged dye solution (resazurin) from dark blue to pink or orange [5,6,7,8,9]. As a result of permanent or partial damage to the bacterial membrane, the dye penetrates only those cells that are dead. Then, as a result of the toxic action of the compounds, there is a loss of potential between the outside and inside of the bacterial membrane. The entire content of the cell is stained with all its components (Appendix A). In our research, we take into account the antibacterial activity of the analyzed lactones as potential drugs, in relation to commonly used antibiotics such as, e.g., Syntarpen, the active ingredient of which is cloxacillin. (Appendix A). The research was carried out with the use of various structural analogs of lactones (Figure 1, Figure 3, and Table 1), containing two aromatic rings linked by different functional groups, such as (methyl) Me, chlorine and (Cl), bromine (B), hydrogen (H) or fluorine (FA) at the position of the R1, R2 or R3 substituent, in their basic structure. In the initial stage of the research, the influence of chloride, bromide, iodide, and fluoride atoms attached to lactone ring (compounds **5a**, **5b**, **5c**, **5d**, **6a**, **6b**, **6c**, **6d**, and **6f**) was investigated. Second set of compounds comprised the molecules containing halogen atoms directly connected to lactone and aromatic group (compounds **5e**, **5f**, **5g**, **5h**, **5i**, and **5j**). Compounds **5a**–**6f** were marked with numbers 1–16 (Table 1). Additionally, as a reference compound, coumarin derivative 17 was used as a reference [5].

The toxicity on bacterial cells was analyzed as per previous studies according to [5]. The best MIC and MBC values among all 17 analyzed compounds were obtained for compounds marked in our study as (**5e–5j** 0.4–0.8 µg/mL for MIC values and 16–37 µg/mL for MBC in strains R2, R3, and R4) (Figure 4 and Figure 5). The antimicrobial activity of these lactones was determined on the basis of the MBC values for the analyzed compounds, which ranged from 30 to 40 µg/mL (Figure 5). The indicator showed an increase in the value of about 160–270 µg/mL in *E. coli* R2–R4 strains compared to the K12 strain, where in, due to the lack of LPS in its structure (Figure 4, Figure 5, and Figure 6), all analyzed MIC and MBC tests values were at a very low level, slightly above zero ((0.15–0.18 µg/mL for the MIC value and 1–3 µg/mL for the MBC) (Figure 4, Figure 5, and Figure 6) and (0–25 µg/mL for the MBC/MIC) (Figure 6)).

On the first MIC plate (Appendix A) where the K12 strain was analyzed, the compounds showed a visible color change at a 10^−6^ dilution, which corresponds to a MIC value of 0.003125 µg/mL^−1^ in all analyzed reactions. On the second plate (panel B), on which strains R2 were used, color changes appeared in all analyzed compounds (Appendix A). A visible color change was observed already at 10^−3^ dilution for compounds marked as 5–10 (**5e–5j**)-having the R3 -fluorine substituent. The MIC values for these compounds were 0.025 µg/mL^−1^. For the remaining analyzed compounds in the R2 shell, the color change was visible at a dilution of 10^−5^, which corresponds to a MIC value of 0.00625 µg/mL^−1^, color change slightly less intense than in compounds containing fluorine. This proves the initiation of the process of destroying the bacterial membrane and the LPS contained in it by the analyzed compounds in the R2 skeleton. With the increase in length of the LPS, the amount of membrane damage began to increase. What we observed exactly in the frames of R3 and R4. The interactions between the analyzed strains and compounds are presented in Figure 3, Figure 4, and Figure 5 and in Appendix A. Increased values for all the analyzed compounds were also observed in both R3 and R4 skins, however, similarly to strain R2, MIC values were observed for five compounds containing fluorine in the R3 substituent, marked as 5–10 (**5e**-**5j**) (Figure 1 and Table 1). A visible color change was already observed at a dilution of 10^−3^ for strain R3 and 10^−2^ in strain R4, which corresponds to MIC values of 0.025 and 0.05 µg/mL^−1^, respectively. The highest antibacterial activity was observed for the strain R4> R3> R2 (Figure 4, Figure 5, Figure 6 and Appendix A). The R4 strain was probably the most sensitive compared to the other strains (Figure 4, Figure 5, and Figure 6). In all the analyzed cases, an appropriate plate was used, with the observed MBC/MIC values being approximately 350 times higher than the MIC (Figure 6) relative to the K12 control strain and statistically significant as shown in Table 2.

### 3.3. Modification of Plasmid DNA Isolated from E. coli R2–R4 Strains with Tested Lactones

Based on the analysis of the MIC and MBC toxicity tests in the analyzed K12 and R2–R4 *E. coli* cells (Appendix A), it was decided to modify the DNA isolated from the analyzed bacterial strains with all 17 analyzed lactones. On the basis of the obtained results of the modification, we observed that in the R4 strain, there was a change in the structure of the analyzed bacterial DNA by spontaneous change of the ccc, linear, and oc forms in relation to each other and the formation of concatamers probably responsible for bacterial DNA topoisomerases (Figure 7 and Appendix A). In the bacterial DNA obtained and modified from other strains, such visible and significant changes in the structure were not observed. However, for further analysis, we used bacterial DNA isolated from all model *E. coli* strains. Since the effect of the Fpg protein was most pronounced in the R4 strain, we provided its example as a model and analyzed it by Tukey’s test.

The obtained DNA was digested with the Fpg-repair glycosylase protein with a broad spectrum of activity, which was analyzed in our previous studies [5,6,7,8,9]. After treatment with Fpg glycosase, we observed clearly visible damage in the topological changes of plasmid DNA forms, i.e., the “ccc”, linear form, and the “oc”, which were totally destroyed and only visible as scattered bands (see Appendix A). In the modified plasmids, we see significant changes between the control and the modified lactones in the electrophoretic images (Appendix A). This may indicate completely new substrates for the Fpg protein itself. Approximately 3–4% or more of oxidative damage was identified in the plasmid DNA after digestion with the Fpg protein (Appendix A), which may indicate that the analyzed compounds are new substrates for this protein. This would negate the fact that the lactones used induce a very high oxidative stress in the bacterial cell, causing strong modifications of DNA bases and disruption of their spatial structure by bacterial topoisomerases in living cells. In the case of plasmids unmodified with selected compounds from strains K12 and R2–R4, three traditional forms were observed: very weak α, linear, and ccc. Similarly to α-amidoamides, coumarin derivatives, 1–2 diaryl alcohol, and ionic liquids [5,6,7,8,9], they can damage plasmid DNA (strongly changing the topological forms of plasmids, even leading to their complete destruction). We can see a similar effect on the example of the compounds we use, e.g., in selected lactones numbered 5–10 (**5e**–**5j**), which evidently form concatamers visible in DNA isolated from strain R4. Moreover, the length of the bacterial lipopolysaccharide may influence the microbiological activity of the analyzed strains and the compounds used on them. The length of the alkyl chain and the structure of all analyzed lactones had a significant effect on the toxicity of the *E. coli* R-type model strains (based on the analysis of the dilution values in the MIC and MBC tests. The analyzed relationships were analyzed with the Tukey’s test and were statistically significant at *p* <0.05 * (Table 2). The results of lactone-modified bacterial DNA in the selected *E. coli* strain R4 are shown in Appendix A.

The next step in our research was the use of δ-lactones as a function of commonly used antibiotics: ciprofloxacin, bleomycin, and cloxacillin a component of the general drug product Syntarpen. Cloxacillin is a beta-lactam antibiotic used to treat bacterial infections, recommended mainly in skin and soft tissue infections, lower respiratory tract infections and osteomyelitis [50,51,52,53,54,55,56,57,58,59,60,61,62,63,64]. Ciprofloxacin is an organic chemical compound, a chemotherapeutic agent from the second-generation quinolone group, with a bactericidal effect, showing its activity by disrupting bacterial DNA replication, inhibiting bacterial DNA topoisomerase and DNA gyrase. It is the most potent drug among fluoroquinolones. Ciprofloxacin is particularly active against Gram-negative bacteria such as *Hemophilus influenzae*, *Neisseria gonorrhoeae*, *Salmonella, Shigella*, *Pseudomonas aeruginosa*, *Escherichia coli*, *Staphylococcus aureus*, *Mycobacterium tuberculosis*, *Enterobacteriaceae*, *Bartonella quintana*,), it shows variable activity towards others [50,51,52,53,54,55,56,57,58,59,60,61,62,63,64].

Bleomycin (Latin Bleomycinum) is a glycopeptide antibiotic obtained from the *Streptomyces verticillus* strain. It is a mixture of polypeptide compounds with cytostatic activity, mainly bleomycin A2 and B2 [50,51,52,53,54,55,56,57,58,59,60,61,62,63,64]. It is a cytostatic antibiotic causing damage to the DNA strand as a result of the binding of the bleomycin–iron ion complex to it, which results in the cleavage of the DNA strand and inhibition of the cell cycle in the G2 and S phase, which in turn leads to the death of the neoplastic cell [50,51,52,53,54,55,56,57,58,59,60,61,62,63,64].

In our previous experiments [6,7], we used a specific range of antibiotics, but here, we used only three antibiotics in the research system due to the fact that we observed similar effects in our earlier works, we wanted to observe its further action on this specific group of compounds, which are lactones [6,7] (Figure 8).

The experimental setup was similar for the lactones used in the MIC and MBC assays, both in terms of quantity and concentration (Figure 9 and Figure 10), and for plasmids isolated from bacteria and modified with these antibiotics and digested with Fpg protein. In all analyzed R-type strains, using the MIC and MBC tests, a color change was observed for the analyzed antibiotics used at a dilution of 10–3 (Appendix A), which corresponds to 0.025 µg/mL^−1^ in the analyzed MIC (Appendix A). It was also shown that the R4 strain, having the longest LPS, interacts with all the active groups contained in the antibiotics. The obtained results were statistically significant at *p* < 0.05. MIC values were similar to those in the R4 strain in the analyzed lactones, especially in compounds 5–10 (**5e–5j**), which proves that these compounds can potentially be used as a replacement for the used antibiotics.

When the bacterial DNA was isolated from all model strains and “reacted” with the antibiotic cloxacillin after digestion with the Fpg protein, the cc, linear, and α forms were observed in various proportions (Appendix A). The greatest damage recognized after digestion with Fpg protein was observed after modification with 5–10 (**5e–5j**) compounds containing fluorine in the R3 substituent. They were the most reactive, and their modification with an antibiotic constituted additional substrates that were recognized by the Fpg protein.

In addition, in bacterial DNA isolated from the R4 strain after treatment with the antibiotic cloxacillin and additionally with the Fpg protein, two additional forms appeared over the ccc and oc forms, which prove the formation of concatamers (additional looped structures in DNA). This effect was especially visible after modification with compounds **6–9** containing fluorine in the substituent R3. In the remaining samples digested with the same antibiotic and not digested with glycosylase, the ratio of all three forms remained the same. After digestion of these samples, streaks appeared as a result of digestion with the Fpg protein recognizing plasmid oxidative damage, visible as “smir”. The intensity of which varied depending on the lactone used. In samples 1–5 and 11–17, the digestion effect with Fpg protein was comparable. After digestion with the Fpg protein in all analyzed cases in the model R4 strain after treatment with cloxacillin, the proportion of topological forms of the analyzed plasmids was disturbed and was visible as a very weak form of ccc along with a barely visible linear form, while the form oc migrated well below the value of both previous forms and was fuzzy. This shows a very strong influence of this antibiotic on the genetic material contained in the cell itself and the destruction of the bacterial membrane in the form of a color change visible in the analyzed MIC and MBC tests (Appendix A).

The highest values of observed damage in bacterial DNA were observed for lactones marked with numbers 5–10, and for samples 6–7 modified with the antibiotic cloxacillin in the R4 strain, the values were the highest; R4> R2> R3> K12. The values of oxidative damage in plasmid DNA after digestion with Fpg protein and treatment with cloxacillin were almost two times lower than after traditional modification of bacterial DNA with the analyzed lactones and ranged from 2.5% to 3.0%. The values of oxidative damage in bacterial DNA recognized by the Fpg protein as “smear” after modification with lactones prove that they damage the DNA of the bacterial cell more than the analyzed antibiotics (disrupted forms of bacterial DNA in the analyzed material), which indicates the toxicity of the analyzed lactones to bacterial DNA, i.e., it strongly damages them by modifying the components of the bacterial membrane and the LPS contained in it.

The endogenous level of Fpg glycosylase in unmodified bacteria is very low, so it is possible that all guanine residues have not been fully repaired in the plasmid DNA and may compete strongly with other repair enzymes in the excision repair system. As a result of the interaction of the analyzed compounds with the bacterial membrane and under the influence of oxidative stress in the cell, bacterial DNA is damaged, which may induce specific bacterial topoisomerases, which allow for structural relaxation and access to modified DNA bases.

This suggests that stabilization of the topoisomerase cleavage complex is essential for the cell as it blocks replication and transcription. In addition, the secondary stability effect of 8oxoG and its derivatives, as well as other modifications to oxidized bases such as Fapy Ade or Gua Faps in the genome, can affect the total amount of superhelicated DNA.

## 4. Discussion

Determination of the toxic effect of selected lactones on model strains of *E. coli* containing or not containing LPS as potentially new substances that show a much stronger effect than traditionally used antibiotics is important for the proper determination of action of new drugs [11]. The use of model *E. coli* strains that contain LPS of different lengths in their structure will allow for a more detailed analysis of the mechanism of disintegration of the bacterial cell membrane and changes in the redox potential of its components.

In our research, we observed that all the 17 compounds used on model bacterial cells were highly toxic, especially the five compounds containing fluorine in their structure in the R3 substituent position, which had the highest antibacterial effectiveness. The highest values in both types of MIC and MBC tests and the highest percent of modification identified after digestion with Fpg protein was observed for compounds designated 5–10 (**5e–5j**). This shows that the type of the substituent and the two aromatic rings determine its toxic activity on the bacterial cells. In the analyzed lactones, which contain various types of substituents in their structure, we observed a significant increase in cytotoxicity in relation to the analyzed model strains of *E. coli* according to R2 < R3 < R4 orders differing in LPS length. It is known from the literature that the analyzed strains can cause diseases related to cardiovascular and digestive system disorders, often leading to the development of various organ dysfunctions and, consequently, to the development of various types of neoplasms [5,6,7,8,9].

The potential toxicity of the selected lactones to the analyzed bacterial cells was high, and the highest values were recorded for the strains R2 and R4 against the strains R3 and K12, similar to other studies with additional compounds [5,6,7,8,9]. For compounds 1–17 containing functional groups consisting of cyclic esters formed as a result of intramolecular esterification of hydroxy acids for the analyzed strains, lactones with different functional groups (Figure 1 and Table 1) were more effective in the R strains than in the K12 strains. The effect of the interaction of the analyzed compounds with the bacterial membrane was very similar to the interaction of ionic liquids containing quaternary ammonium surfactants [8,9].

Changes in the activity of bacterial cells due to the given compounds probably result from the rearrangement of the polarity of the components of the bacterial membrane containing LPS as a result of interaction with lactones exhibiting strong cytotoxicity, inducing oxidative stress in the bacterial cell, leading to its damage and biochemical decomposition of genetic material [21,22,23,24,25,26,27,28,29,30,31,32,33,34,35,36,37,38,39,40].

The obtained results constitute the basis for the continuation of further research on other pathogenic bacterial strains associated with diseases of the vital systems related to the basic functioning of the human body. They will also allow to identify potential mechanisms of degradation in cell membranes through newly synthesized substances as new precursors of known and commonly used antibiotics [21,22,23].

The analysis of the toxicity of lactones used in our research shows that it is strongly related to the length of the LPS in the analyzed types of bacteria R2–R4. In addition to the MIC and MBC tests, the digestion analysis of the modified plasmids isolated from strains K12 and R2, R3, and R4 by Fpg protein with N-glycosylase/AP lyase activity was performed [5,6,7,8,9,10,11,51,52,53].

Percentage of damage to plasmids modified by lactones was determined from the changes in the topological forms of ccc, linear, and oc after treatment with the Fpg protein. From literature data, Fpg protein is known to have a wide range for recognizing and eliminating oxidized and alkylated bases that have been modified by ROS or RNS. Fpg protein is now recognized by many international laboratories as a sensitive ‘marker’ of oxidized bases formed in bacterial cells under the influence of oxidative stress caused by internal and external factors. It is assumed that the amount of identified base damage above 3–4% in a single or double strand of DNA caused by oxidation or alkylation by the enzyme Fpg is a very important indicator of the degree of guanine or adenine modification in the analyzed genetic material [51,52,53].

In our research, protein-Fpg recognizes modifications introduced by lactones to plasmid DNA, especially in the case of five compounds selected from all tested compounds. Visible changes between topological forms and the emerging so-called smearing of the bands after digestion of the modified plasmids with the protein results from providing the enzyme with new potential substrates for its repair activity under the influence of specific glycosylases used [51,52,53].

The results suggest that the selected compounds modify the bacterial DNA isolated from the model K12 and R strains and are recognized by the Fpg glycosylase (Appendix A). The compounds **5e–5j** are the most effective. This means that in the future, certain lactones could be designed as new potential substitutes for antibiotics with very similar chemical structures but stronger activity against all bacterial strains. According to literature data, frequent chemotherapy with antibiotics resulted in immunization of many bacterial pathogens [29]. Therefore, we additionally tested cloxacillin, a component of the drug product Syntarpen, which is administered to patients in the case of acute gastrointestinal or cardiovascular inflammation by bacteria of the genus *Staphylococcus aureus* [30]. The antibiotic has a very broad spectrum of activity against Gram-negative bacteria, including *E. coli* [31], as shown in appropriate MIC plates (Appendix A). After treatment of the bacterial DNA with the selected antibiotic and additional digestion with the Fpg protein, the highest level of damage was observed for the strains R4 > R2 > R3 > K12. These values were twice as high with the respective lactones compared to the treatment with cloxacillin with bacterial DNA in all analyzed strains, especially in the R4 strain (Appendix A). Therefore, it is still advisable to look for compounds with a similar structure, but stronger biological and chemical properties, which would be more toxic to bacterial cells.

On the basis of the applied MIC and MBC tests, it is possible to design lactones with a very high toxicity to Gram-negative bacterial cells, similarly to the previously analyzed α-amidoamides, coumarin derivatives, 1,2-diaryl alcohols, and ionic liquids [5,6,7,8,9]. The practical application of the analyzed compounds will allow their use in the future as potential new antibiotics that will be more toxic and effective against all the tested model strains of *E. coli* bacteria. A multi-component reaction was used to discover novel properties of the selected five candidate compounds as shown in Table 1. The studies showed that the samples subjected to Fpg digestion showed “high smear” and rearrangement of the topological forms of the plasmid, especially in the case of the bacterial DNA isolated from the R4 strain (Appendix A). This means that the enzymes recognized damage caused by modifications with antibiotics that are “toxic” to DNA in the same way as in the case of the bacterial DNA modified with selected lactones. The structure of Fpg, which determines its dual activity as protein glycosylase and AP lyase, consists of two domains. N-terminal domain containing the active site within the first 72 amino acid residues and the secondary amino group containing Pro1 and Glu2 as well as Lys56 and Lys154. Pro1 and Glu2 are essential for the activity of the Fpg enzyme in the recognition of modified bases. The C-terminal domain contains the motif herpin-lix-harpin-helix (HhH) and is involved in DNA binding. Functional homologues of formamidopyrimidine DNA glycosylase have now been identified in *Saccharomyces cerevisiae* (yOGG1) and humans (hOGG1) [51,52,53].

In the bacterial DNA isolated from the R4 strain digested with lactone-modified Fpg protein and after modification with antibiotics (including cloxacillin), we observed the disappearance of “ccc” forms and the appearance of a single strand migrating similarly, but slightly slower than the “oc” form (see Appendix A). The migration effect of the bands in both experiments was similar to each other and confirm that selected lactones can replace antibiotics in the future, enhancing their biological functions by affecting the bacterial cell membrane. This is indicated by the results of digestion with the Fpg protein (Appendix A), which forms a strong complex in the form of concatamers with modified DNA plasmids. This complex is strong enough and stable in electrophoresis. He suggests that the Fpg protein was covalently linked to DNA via amino acids or high-molecular-weight protein fragments and recognizes some induced DNA damage following modification by lactones and the cloxacillin-type antibiotic.

This suggests that protein–DNA cross-linking by δ-lactone groups of the resulting adducts may be the proper mechanism for their formation, but the extent of this may vary depending on the protein. Repair of the analyzed three topological forms of plasmids in DNA in the case of the compounds used is carried out by the BER system, which includes Fpg glycosylase. Then, the aforementioned DNA–protein complexes arise, visible as blurred bands or compact forms in the form of concatamers, which move slower during agarose gel electrophoresis. We postulate that the interaction between DNA and lactones and antibiotics occurs through a covalent bond between lactone groups and other groups present as R1, R2, and R3 substituents with the presence of double aromatic rings that can arise in single-stranded plasmid DNA. Each band visible on the gel represents the nucleotide positions at which strand damage or breakage was induced by digestion with DNA glycosylase/AP lyase. Analysis of the mapped sites recognized by the Fpg enzyme suggests that all DNA residues are linked to them as modified oxidized bases that may correspond to the 8-oxoguanine sites in the template. Probably, the spatial rearrangement of bases and the selection of an appropriately modified base by the analyzed compounds are the result of the rearrangement of the chain, which is efficiently digested by Fpg glycosase, which is a new substrate for them. The addition of specific groups of lactones causes thermodynamic destabilization of a single helix on cytosine and guanine [32,33] by loosening the compact structure of CG-rich regions that were more easily bypassed by Fpg enzymes, allowing for the identification of damaged cytosine residues and the participation of specific topoisomerases.

The nature of the DNA damage mechanism recognized by Fpg glycosylase results from its construction and requires further elucidation [51,52,53].

Similar to our previous works, the mechanism of recognizing damaged (oxidized bases) by the Fpg enzyme is very similar as described in [51,52,53]. In general, the R4 strains showed a higher sensitivity than the R2 and R3 strains (Figure 3, Figure 4, Figure 5, Figure 6, Figure 7, and Figure 8 and Appendix A). The analyzed LPS-containing bacterial strains showed increased sensitivity to the toxic effects of lactones of varying intensity (compared to the control strain K12). The R4 strain, due to the longest LPS length, was probably the most sensitive. Studies of cytotoxic activity on bacterial cells of the newly synthesized compounds will allow for a better selection of type of substituents, leading to the creation of drugs with the best biological parameters and microbial activity for the analyzed bacterial cells. The newly obtained compounds are a new alternative as potential innovative substitutes for antibiotics due to their specific structure related to commonly used drugs. The presence of fluorine ions as a substituent in the analyzed lactones caused significant damage to bacterial DNA as measured by MIC and MBC tests as well as by digestion with Fpg protein. This protein is a marker of the oxidative stress generated during lipid peroxidation, which is believed to be an important mechanism of fluorosis. A close relationship between the toxicity of fluoride and oxidative stress was also noted in cultured cells [53,54,55] and experimental animals [56]. Studies have shown that an excess of fluoride can cause DNA damage, induce apoptosis, and change the cell cycle [55,56,57,58]. Jeng et al. [59] investigated the effect of sodium fluoride on human oral mucosa fibroblasts and found that sodium fluoride is toxic in vitro by inhibiting protein synthesis, disrupting mitochondrial respiratory function, and destroying cellular ATP through oxidative phosphorylation. Fluorine is one of the halogens and the most electronegative element leading to a greater polarization of the C-F bond, which is less covalent and has more electrostatic/dipole [60,61]. Due to the small size of fluorine in relation to the hydrogen atom, it can very closely mimic hydrogen in its non-fluorinated analogs, allowing the fluorinated precursor to fit spatially to the analyzed enzyme receptor, thus increasing its biological activity against the analyzed K12 and R2–R4 bacterial cells [60,61]. Although the carbon–fluorine bond is stronger and more thermally stable than the carbon–hydrogen bond, fluorine is still a better group than hydrogen alone [60,61]. Hence, fluorine, as a substituent, results in improved activity in lactones, making the compounds containing it more toxic in their effects on bacterial cells than the other analyzed lactones presented in Table 1.

Fluorine is the most active non-metal with the highest electronegativity, forming compounds with most other elements (even with noble gases—krypton, xenon, and radon). Therefore, it creates intra- and intermolecular interactions in ligands and supramolecular complexes, studied by NMR spectroscopy. Elemental fluoride as well as fluoride ions are highly toxic. It disrupts enzymatic processes in cells, inhibiting tissue respiration, the transformation of carbohydrates and lipids, and the synthesis of hormones. Fluoride itself and some of its compounds are corrosive, causing deep necrosis. Free fluorine has a characteristic irritating smell, which is noticeable even at a concentration of 20 ppm [61,62,63,64,65,66].

It is believed that fluoride’s ability to induce oxidative stress by generating free radicals and reducing the efficiency of the enzyme antioxidant system plays an important role in the mechanism of fluoride’s toxicity. Fluoride as a pro-oxidant induces increased free radical processes that lead to damage of lipids, proteins, DNA in the kidneys, liver, brain, and blood [67,68,69]. Depending on the dose and exposure time, it may be an inhibitor of antioxidant enzymes such as superoxide dismutase, catalase and glutathione peroxidase [70,71]. Due to their biochemical and physical properties, fluoride ions easily penetrate cell membranes, penetrating both hard and soft organ tissues. The toxicity of fluoride depends on the dose and the duration of its exposure. Chronic or acute exposure to fluoride may occur due to the presence of fluoride in the diet or as a result of environmental contamination with fluoride compounds. The result is dental and skeletal fluorosis. In the first stage, white spots appear on the teeth, and then, the enamel darkens and cracks [72]. In the metabolism of osteoblasts and osteoclasts, changes occur that lead to abnormal bone mineralization. Bones become more fragile and ligaments lose their elasticity [73]. The organs most involved in detoxifying and eliminating fluoride from the body are the liver and kidneys. The main route of fluoride removal from the body is through urine excretion [74]. Excessive supply of fluoride causes degeneration and insufficiency of the renal tubular epithelium, hypertrophy, and atrophy of the renal parenchyma [75,76]. The liver exhibits inflammation and even necrotic changes, inhibition of protein synthesis, and disturbance of its detoxification functions [75,77,78]. The phenomenon of fluoride hyperglycemia is also known, resulting from, inter alia, inhibition of insulin secretion by pancreatic β cells and disturbed secretion of thyroid hormones [79,80]. Moreover, fluoride is highly neurotoxic. It causes neurodegenerative changes in the cerebellum, hippocampus, and cortex [81]. This leads to an inhibition of the synthesis of neurotransmitters and a reduction in the number of their receptors in the brain [82]. Prolonged exposure to fluoride results in damage to the reproductive organs and infertility, increasing the percentage of miscarriages, and is very toxic to the fetus [83,84]. It was also found that fluoride ions reduce the activity of the sodium–proton exchanger. Fluoride has a similar unfavorable effect on energy transformations in cells [83,85]. In the case of chronic or acute fluoride poisoning, disturbances in the mineral balance appear. It also forms toxic complexes with calcium, magnesium, phosphorus, and aluminum [86,87]. Aluminum–fluoride complexes formed in aqueous solutions containing fluorine and trace amounts of aluminum ions show toxic properties at the cellular level and are considered to be one of the factors damaging the central nervous system [88]. Therefore, knowing about the negative role of fluoride as an element on eukaryotic cells, we used ions of fluoride in our compounds to see how they would behave towards bacterial cells.

## 5. Conclusions

The conducted research confirms the usefulness of model strains of *E. coli* in terms of the analysis of the toxic effects of various new, innovative compounds, which include lactones, in order to identify new natural substances similar to antibiotics. In our experiments, we found that:δ-lactones with a specific structure, including those containing fluorine in the R3 substituent (Figure 1 and Table 1),Lactones are able to modify all strains of *E. coli* (R2–R4) and their plasmid DNA, and further, they also change the spatial structure of the LPS contained in their cell membrane.We found that among the analyzed model strains of *E. coli*, the R4 type strain was the most sensitive, which is related to the length of its LPS.The interaction of the analyzed lactones with the cell membrane of the K12 strain indicates differences in the O-antigen and the core of the truncated oligosaccharide compared to the analyzed R-type strains, which may play an important role in the cellular response to charged compounds.The toxicity of aromatic groups together with alkyl substituents depends on their interaction with the membrane, which can become involved in cell wall structures and change their hydrophobicity.Changes in the structure of the bacterial membrane and disorders of its integrity may result in changes in the bacterial response to other biologically active compounds, such as antibiotics.Plasmid DNA damage is linked to the structure of the verified compounds, suggesting that the presence of lactones affects LPS bacteria and generates oxidative stress, which we have already observed in our previous studies [5,6,7,8,9].The tested lactones show a different effect in the MIC and MBC tests, which is strongly correlated with the spherical factor of functional groups in the form of substituents with short-chain alkyl in the structure of the analyzed compounds [5,6,7,8,9,10,11]. The results of our new experiments are important for understanding the biological properties of the lactones being tested as a function of potential new antibiotics and their toxic effects on Gram-negative bacteria in the face of increasing drug resistance and epidemic infections.The δ-lactone group plays a key role in the interaction with other biomolecules (enzymes and receptors), which results in the observed biological activity of the analyzed Fpg protein after digestion of bacterial DNA.The δ-lactone compounds containing fluorine in the substituent structure turned out to be the most cytotoxic for bacterial cells in relation to other analyzed lactones.The analyzed lactones are more cytotoxic in the model bacterial cells used for the research than the antibiotics ciprofloxacin, bleomycin, and cloxacillin used. After the modification of the bacterial DNA with the analyzed antibiotics, the degree of damage to the DNA bases recognized by the Fpg protein is much lower than after the use of the analyzed lactones. In addition, the antibiotics used in the MIC and MBC tests show higher values in micrograms per millimeter than the analyzed lactones, which proves their lower effectiveness and cytotoxicity on the bacterial cells used.

## Figures and Tables

**Figure 1 materials-14-02956-f001:**
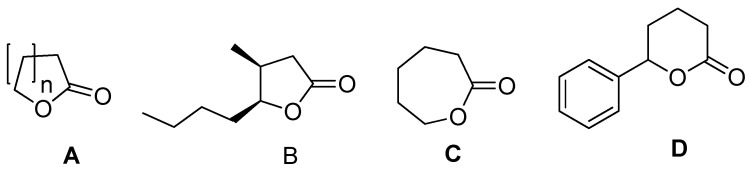
Representative examples of lactone structure: lactone (**A**), whiskey lactone (**B**), ε-Caprolactone (**C**), and 6-phenyl tetrahydro pyran-2-one (**D**).

**Figure 2 materials-14-02956-f002:**
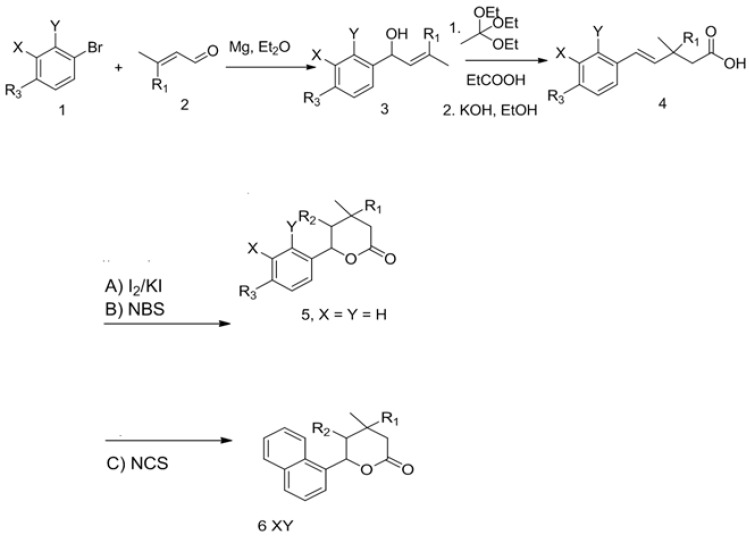
General synthetic method used for the synthesis of target lactones **5** and **6**.

**Figure 3 materials-14-02956-f003:**
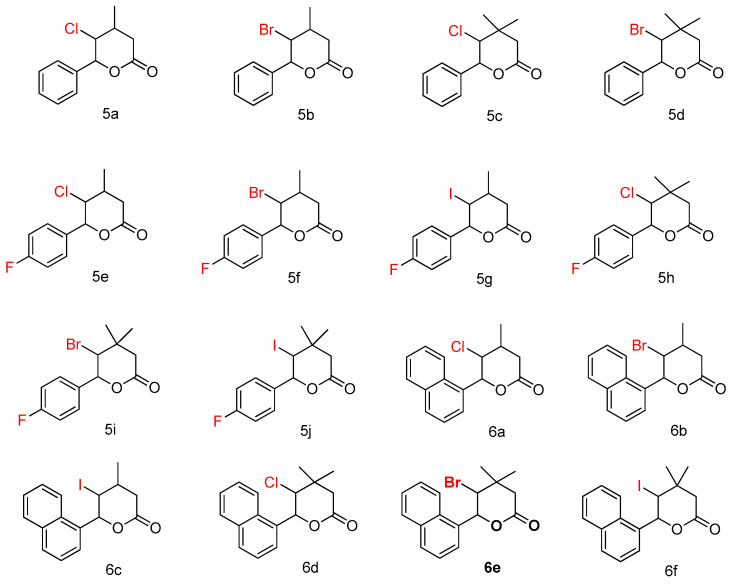
The chemical structure of δ-lactones obtained. The specific substituents that determine the reactivity of the compounds are marked in red.

**Figure 4 materials-14-02956-f004:**
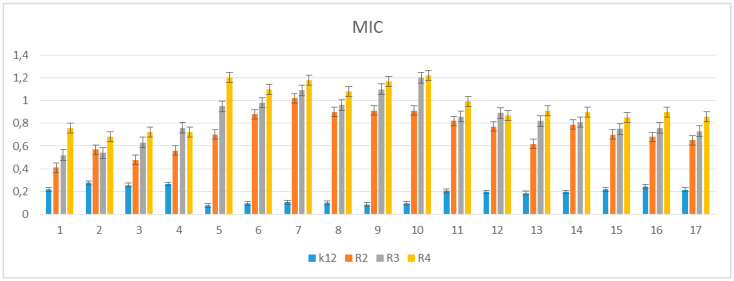
Minimum inhibitory concentration (MIC) of the lactones in model bacterial strains. The *x*-axis compounds **1**–**17** were used sequentially. The *y*-axis shows the MIC value in µg/mL^−1^.

**Figure 5 materials-14-02956-f005:**
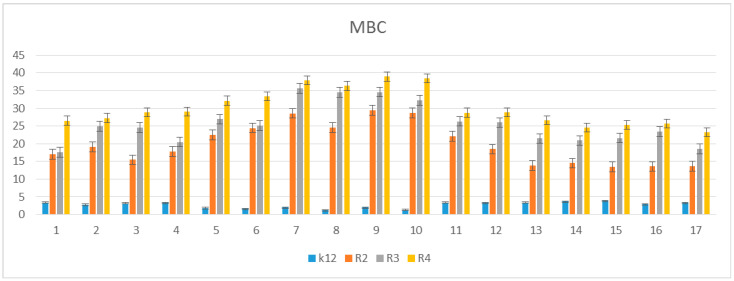
MBC of the lactones in model bacterial strains. The *x*-axis compounds **1**–**17** were used sequentially. The *y*-axis shows the MBC value in µg/mL^−1^.

**Figure 6 materials-14-02956-f006:**
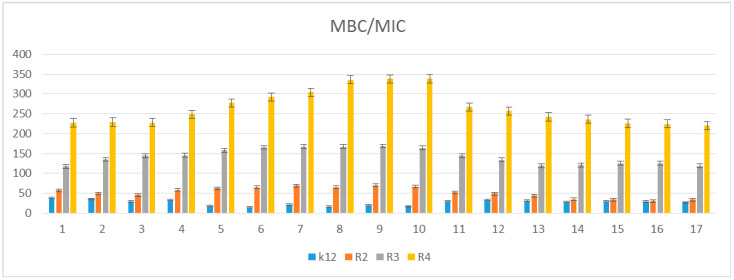
MBC/MIC of the lactones in model bacterial strains. The *x*-axis compounds **1**–**17** were used sequentially. The *y*-axis shows the MBC/MIC value in µg/mL^−1^.

**Figure 7 materials-14-02956-f007:**
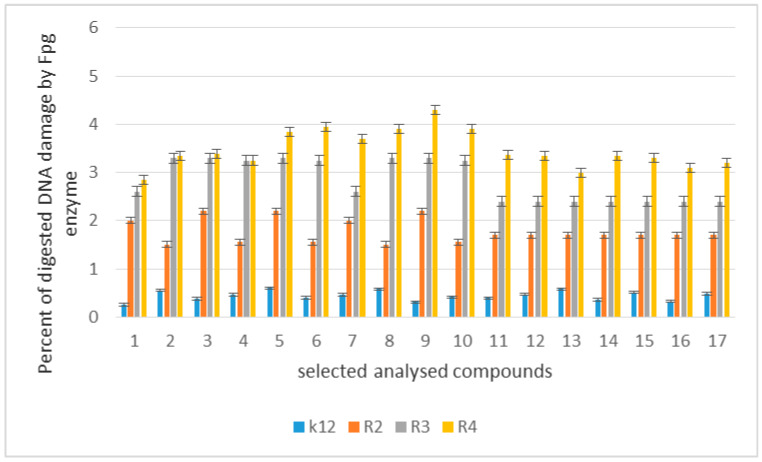
Percent of digested DNA damages recognized by Fpg enzyme (*y*-axis) with control K12 and R2–R4 strains (*x*-axis). The selected compounds **1**–**17** were statistically significant at *p* < 0.05 *.

**Figure 8 materials-14-02956-f008:**
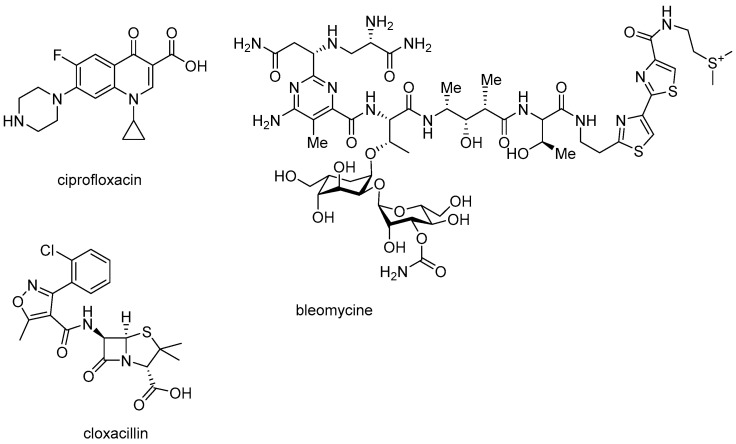
The structure of antibiotics ciprofloxacin, bleomycin, and cloxacillin used as reference compounds.

**Figure 9 materials-14-02956-f009:**
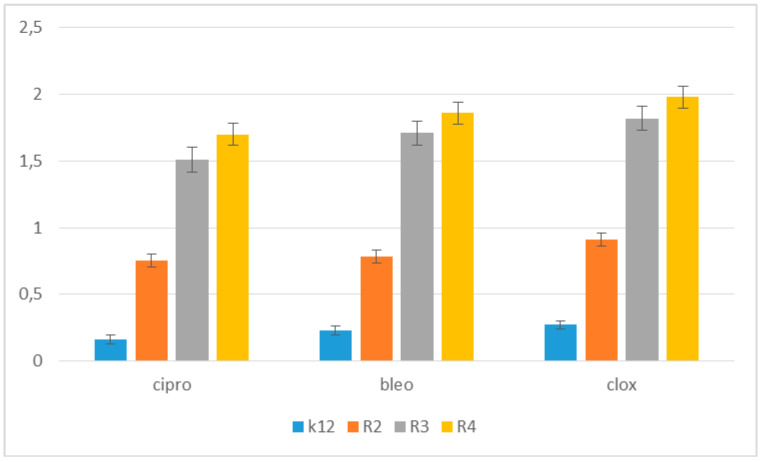
Examples of MIC with model bacterial strains K12, R2, R3, and R4 of the studied antibiotics with ciprofloxacin, bleomycin, and cloxacillin. The *x*-axis features antibiotics used sequentially. The *y*-axis features the MIC value in µg/mL^−1^.

**Figure 10 materials-14-02956-f010:**
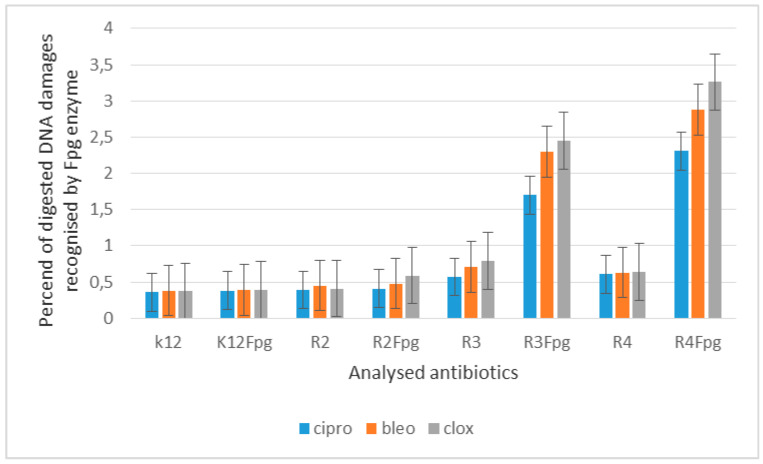
Percentage of bacterial DNA recognized by Fpg enzyme in model bacterial strains after ciprofloxacin, bleomycin, and cloxacillin treatment. The compounds were statistically significant at *p* < 0.05.

**Table 1 materials-14-02956-t001:** Yields of reaction provided for isolated pure products.

Entry	Compound	R1	R2	R3	Yield
1	**5a**	H	Cl	H	84%
2	**5b**	H	Br	H	72%
3	**5c**	Me	Cl	H	85%
4	**5d**	Me	Br	H	59%
5	**5e**	H	Cl	F	81%
6	**5f**	H	Br	F	76%
7	**5g**	H	I	F	37%
8	**5h**	Me	Cl	F	65%
9	**5i**	Me	Br	F	85%
10	**5j**	Me	I	F	60%
11	**6a**	H	Cl	–	82%
12	**6b**	H	Br	–	82%
13	**6c**	H	I	–	80%
14	**6d**	Me	Cl	–	75%
15	**6e**	Me	Br	–	62%
16	**6f**	Me	I	–	93%

**Table 2 materials-14-02956-t002:** Statistical analysis of all analyzed 17 compounds at *p* < 0.05 *, *p* <0.01 **, and *p* <0.001 *** in MIC, MBC, and MBC/MIC tests. Lactones 1–17 were used sequentially.

No of Samples	1	2	3	4	5	6	7	8	9	10	11	12	13	14	15	16	17	Type of Test
K12	*	*	*	*	*	*	***	***	*	**	**	***	**	***	*	***	**	MIC
R2	*	*	*	*	*	*	***	***	*	***	**	***	**	***	*	***	**	MIC
R3	*	*	*	*	*	*	***	***	*	**	**	***	**	***	*	***	**	MIC
R4	*	*	*	*	*	*	***	***	*	**	**	***	**	***	*	***	**	MIC
K12	*	*	*	*	*	*	**	**	*	***	***	**	***	**	*	***	***	MBC
R2	*	*	*	*	*	*	**	**	*	***	***	**	***	**	*	***	***	MBC
R3	*	*	*	*	*	*	**	**	*	***	***	**	***	**	*	***	*	MBC
R4	*	*	*	*	*	*	**	**	*	***	***	**	***	**	*	***	*	MBC
K12	*	*	*	*	*	*	*	*	*	**	**	*	**	*	*	***	*	MBC/MIC
R2	*	*	*	*	*	*	*	*	*	**	**	*	**	*	*	***	*	MBC/MIC
R3	*	*	*	*	*	*	*	*	*	**	**	*	**	*	*	***	*	MBC/MIC
R4	*	*	*	*	*	*	*	*	*	**	**	*	**	*	*	***	*	MBC/MIC

## Data Availability

On request to those who are interested.

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
