# Peer review of "δ-Lactones—A New Class of Compounds That Are Toxic to E. coli K12 and R2–R4 Strains"

_materials, 2021, doi:10.3390/ma14112956_

Round 1

Reviewer 1 Report

Introduction

  • Line 49/50: I do not understand the sentence- lactones are, as clarified in the first sentence cyclic esters, so what is this sentence supposed to say?

  • Figure 1: Labels A – D should be described in the figure legend (line 55).
  • Lines 63-73: Describes an interesting role of lactones on our sensory receptors, but it is not relevant to the objectives or results of the study. Can be removed from introduction.
  • Line76: these are multiple publications. In which context/ which characteristics of E.coli were described. What is the context? The coumarins?

  • Line 77-78: is this connected to e.coli or coumarines?

  • Line 74-81: Neither the specific properties of coumarins nor Alzheimer are relevant in this context.

  • Line 83-87: not relevant

  • Line 88: Fix English expression.
  • Line 95 – 99 are two sentences basically saying the same thing and refer to the same references [18-24]. These can be condensed into one sentence to read better.
  • Line 104-105: actinomycins and Actinomycetes antibioticus is now more commonly referred to as streptomycins and Streptomyces antibioticus, respectively. Please change in the text, or rewrite to “actinomycins produced by Actinomyces antibioticus cultures (also known as streptomycins and Streptomyces antibioticus, respectively).”
  • 106-107 “this antibiotic’_-which one?, in the sentences before you are talking about the cytostatic drug actinomycin d

  • Please include a summary of the aims/hypothesis or purpose of the study in the introduction.
  • even though you elaborate a lot about the properties of lactones, it is not clearly shown why you are investigating the chosen structure. Further more there is no deduction about why you focused on E. Coli.

Materials and Methods

  • Section 2.3: What do “alcohols 3” and “carboxylic acid 4a-c” refer to? I imagine it is referring to Figure 2, but this is undefined.
  • Section 2.3.1 – 2.3.16: This level of detailed chemical property information is appreciated, but it is tedious to read and for comparing compounds. Perhaps it would be better to include in a table (S1) in the supplementary material.
  • You have two section 2.3’s. Fix.
  • Section 2.7: This needs to be rewritten and some analysis be redone. Remove “ANOVA” or “analysis of variance” – they are the same thing, no need to say both. Why did you describe using both “post hoc LSD” (i.e. Fisher’s Least Significant Difference (LSD) test) and Tukey’s multiple comparison when you only show the Tukey’s multiple comparison test results in Table 2? Fisher’s LSD test does *not* perform multiple comparisons for the data, but multiple individual t-tests with pooled SD’s. So, there is no correction for multiple comparisons and with so many t-tests being performed, as you have 17 groups, you are greatly increasing your chance for type I or II errors. There is no justification for Fisher’s LSD test, just perform the Tukey’s multiple comparisons test instead. R values are Spearman correlation coefficients, not means. What is the number of experimental replicates represented in each figure?
  • Cytotoxicity study needed (cells)

  • You do not cite appropriate literature (i.e. the original papers) for methods, instead you use too many self-citations
  • If you do not write down the used methods (generally you at least give a brief description) you have to quote a paper which explains it correctly and in detail. i.e. Of the cited papers, only Kowalczyk, P.; Borkowski, A.; Czerwonka, G.; Cłapa, T.; Cieśla, J.; Misiewicz, A.; Borowiec, M.; Szala, M. The microbial toxicity of 853 quaternary ammonium ionic liquids is dependent on the type of lipopolysaccharide. J. Mol. Liq. 2018, 266, 540–547 and Borkowski, A.; Kowalczyk, P.; Czerwonka, G.; Cieśla, J.; Cłapa, T.; Misiewicz, A.; Szala, M.; Drabik, M. Interaction of quaternary 855 ammonium ionic liquids with bacterial membranes—Studies with Escherichia coli R1–R4-type lipopolysaccharides. Actually describes the methods of MIC/MBCI was not willing to look this up for every method/reference, but expect this to be corrected and will only then be able to evaluate the methods used. If you cite another paper, just cite 1 correct one and not multiple irrelevant ones just to increase amount of quotations for the other papers. It is also not appropriate to just cite papers here to increase your own reference record, which is here obviously the case. This section definitely needs improvement. This also applies to in-text references in the result section (ie line 494)

Results

  • Lines 364 – 386 and 585 – 598 do not fit the results section as the Journal describes. It is more appropriate in the relevant part of the discussion section.
  • Figure 2: Simplify by removing “XY = -CH=CH-CH=CH-“ – X and Y are no longer labelled on compound 6 so this is redundant.
  • Line 398 - 400: When referring to Table 1, you should refer to figure 3, and when referring to Figure 3, you should be referring to Table 1. Switch them. “High yields” is subjective, please specify which high-yield single delta-lactones you are referring to.
  • Figure S1 and S4: concentrations of compounds used for the MIC and MBC assays are not stated.
  • Correct the MIC and MBC units of concentration. You state mg/mL (milligrams/mL) in the figure texts and µg/mL in the main text. Regardless, these concentration units should all be converted to molarity for better comparison between compounds and compared to ciprofloxsacin, bleomycin and cloxacillin (Figure 9)., since compounds 6a – 6f are higher molecular weight than 5a-5j.
  • Line 431 – 434: This sentence is not expressed correctly.
  • Line 437: what are the differences in these strains? How many replicates? How was the dilution? Is there any standard AB as comparison/positive control?
  • Line 439 – 440 and 451: Please remove mentions of concentration for MBC/MIC, it cannot be expressed as a concentration, it is purely a ratio.
  • Line 452 - 476: Correct “10-6 dilution”, “mg/ml-1”, “dilution of 10-5” and the like in this paragraph and the rest of the main text.
  • Line 474: remove “a 48-well plate was used”.
  • Table 2: Indicate that these results are based on Tukey’s multiple comparison test, like you say in line 517 – 518. It’s not immediately obvious if this was the Tukey’s test or Fisher’s LSD test.
  • Move “covalently closed circle” and “open circle” from lines 496-497 to line 485 where “ccc” and “oc” are first mentioned.
  • Line 576: remove “she” and change “proves” to something like “shows” or “exhibits” etc.
  • Line 587 – 588: “… antibiotic. smirów (disrupted forms of bacterial DNA in the analyzed material. This shows that…” What is this?
  • Line 593: Change to “base excision repair system.”
  • Improve graphs, especially label on axis, but also description

Discussion

  • Lines 604 – 608 are expressed poorly. Improve writing.
  • Line 611: “Highest lues in both types of MIC and MBC tests”? Don’t you mean lowest? Also, you need to check if this is true after changing these values to molarity concentration units.
  • Lines 617 – 620: References [5-9] do not support this statement. Make the appropriate citations of the correct articles to support this.
  • Line 652: remove “and strength”.
  • Lines 669 – 670: correction, “aureus”, remove “48-well”.
  • Line 680 - 682: saying “all strains of pathogenic bacteria” is far too overreaching since you only tested it on coli, make a more conservative statement. Remove quotation marks around “antibiotics”.
  • Line 696: italicise Sacchromyces cerevisiae.
  • Line 708: “He”?
  • Remove sentence in Line 738 – 739: “Our research shows that both… all analyzed compounds (Fig. 4-6).”
  • Line 747 – 766: I disagree with this argument about Fluorine ions. Having fluoryl groups instead of H atoms can make a huge difference to the activity of the drug and does not necessarily mean there is reactions of lactones producing free fluoride ions. This would need to be shown by fluoride determination assays after lactone treatment of the coli.
  • It does not really become clear how this lactones can be used as potential antibiotic treatments. Only one bacteria species was looked at and only in a very basic way without any toxicity studies in (human) cells.

Conclusions

  • Conclusions need to be more succinct and directly relevant to the results in this paper.

Author Response

Reviewer 1

Thank you very much for the substantive suggestions that will contribute to increasing the quality and scientific value of our manuscript.

Introduction

  • Line 49/50: I do not understand the sentence- lactones are, as clarified in the first sentence cyclic esters, so what is this sentence supposed to say?

             This sentence highlighted the important feature of lactones structure associated with the presence of intramolecular ester group. Analogous compound with the extramolecular ester group cannot be recognized as lactons but cyclic esters.   

              Figure 1: Labels A – D should be described in the figure legend (line 55).

  • Additional label was added to figure 1 legend.      
  •  
  • Lines 63-73: Describes an interesting role of lactones on our sensory receptors, but it is not relevant to the objectives or results of the study. Can be removed from introduction.
  • Accodring to referee suggestion these sentence was removed from introduction
  •  
  • Line76: these are multiple publications. In which context/ which characteristics of E.coli were described. What is the context? The coumarins?

Accodring to referee suggestion these sentence has been corrected

  • Line 77-78: is this connected to e.coli or coumarines?

Accodring to referee suggestion these sentence has been corrected

  • Line 74-81: Neither the specific properties of coumarins nor Alzheimer are relevant in this context.

Accodring to referee suggestion these sentence has been removed

  • Line 83-87: not relevant

This sentence highlighted the important class of ε- lactones on one particular example. The world production of ε-Caprolactone exceeds the sum of production of other lactons altogether. 

  • Line 88: Fix English expression.
  •  
  • The sentence has been corrected
  •  
  • Line 95 – 99 are two sentences basically saying the same thing and refer to the same references [18-24]. These can be condensed into one sentence to read better.
  • The sentence has been corrected

  • Line 104-105: actinomycins and Actinomycetes antibioticus is now more commonly referred to as streptomycins and Streptomyces antibioticus, respectively. Please change in the text, or rewrite to “actinomycins produced by Actinomyces antibioticus cultures (also known as streptomycins and Streptomyces antibioticus, respectively).”
  • The sentence has been corrected
  •  
  • 106-107 “this antibiotic’_-which one?, in the sentences before you are talking about the cytostatic drug actinomycin
  • The sentence has been corrected
  • Please include a summary of the aims/hypothesis or purpose of the study in the introduction.
  • The sentence has been corrected
  • even though you elaborate a lot about the properties of lactones, it is not clearly shown why you are investigating the chosen structure. Further more there is no deduction about why you focused on E. Coli.
  •  The sentence has been corrected

Materials and Methods

  • Section 2.3: What do “alcohols 3” and “carboxylic acid 4a-c” refer to? I imagine it is referring to Figure 2, but this is undefined.
  • The structures of all compounds are depicted on Figure 2 what I common method in this Journal.

  • Section 2.3.1 – 2.3.16: This level of detailed chemical property information is appreciated, but it is tedious to read and for comparing compounds. Perhaps it would be better to include in a table (S1) in the supplementary material.
  • We have followed general guide for manuscript preparation of Journal.
  • You have two section 2.3’s. Fix.

This two sections has been corrected

  • Section 2.7: This needs to be rewritten and some analysis be redone. Remove “ANOVA” or “analysis of variance” – they are the same thing, no need to say both. Why did you describe using both “post hoc LSD” (i.e. Fisher’s Least Significant Difference (LSD) test) and Tukey’s multiple comparison when you only show the Tukey’s multiple comparison test results in Table 2? Fisher’s LSD test does *not* perform multiple comparisons for the data, but multiple individual t-tests with pooled SD’s. So, there is no correction for multiple comparisons and with so many t-tests being performed, as you have 17 groups, you are greatly increasing your chance for type I or II errors. There is no justification for Fisher’s LSD test, just perform the Tukey’s multiple comparisons test instead. R values are Spearman correlation coefficients, not means. What is the number of experimental replicates represented in each figure?
  • the statistical analyzes described were described in our previous work. The experimental replicates represented in each figure are minimum three.

  • Cytotoxicity study needed (cells)

In chapter 2.4 has been corrected

  • You do not cite appropriate literature (i.e. the original papers) for methods, instead you use too many self-citations
  •  
  • We agree with this statement, but we wanted to present the research problem without using our self-citations
  • If you do not write down the used methods (generally you at least give a brief description) you have to quote a paper which explains it correctly and in detail. i.e. Of the cited papers, only Kowalczyk, P.; Borkowski, A.; Czerwonka, G.; Cłapa, T.; Cieśla, J.; Misiewicz, A.; Borowiec, M.; Szala, M. The microbial toxicity of 853 quaternary ammonium ionic liquids is dependent on the type of lipopolysaccharide. J. Mol. Liq. 2018, 266, 540–547 and Borkowski, A.; Kowalczyk, P.; Czerwonka, G.; Cieśla, J.; Cłapa, T.; Misiewicz, A.; Szala, M.; Drabik, M. Interaction of quaternary 855 ammonium ionic liquids with bacterial membranes—Studies with Escherichia coli R1–R4-type lipopolysaccharides. Actually describes the methods of MIC/MBCI was not willing to look this up for every method/reference, but expect this to be corrected and will only then be able to evaluate the methods used. If you cite another paper, just cite 1 correct one and not multiple irrelevant ones just to increase amount of quotations for the other papers. It is also not appropriate to just cite papers here to increase your own reference record, which is here obviously the case. This section definitely needs improvement. This also applies to in-text references in the result section (ie line 494)

 Accodring to referee suggestion these references in the result section has been corrected

Results

  • Lines 364 – 386 and 585 – 598 do not fit the results section as the Journal describes. It is more appropriate in the relevant part of the discussion section.
  • We want to highlight the importance of our studies which are not that clear to readers not familiar with modern organic chemistry methods. The biological activity of structurally similar lactams were studied intensively due to easy accessibility of target molecules.

  • Figure 2: Simplify by removing “XY = -CH=CH-CH=CH-“ – X and Y are no longer labelled on compound 6 so this is redundant.

The Figure 2 contents was modified according to referee note.

  • Line 398 - 400: When referring to Table 1, you should refer to figure 3, and when referring to Figure 3, you should be referring to Table 1. Switch them. “High yields” is subjective, please specify which high-yield single delta-lactones you are referring to.
  • The sentence was corrected
  •  

  • Figure S1 and S4: concentrations of compounds used for the MIC and MBC assays are not stated.

Has been corrected

  • Correct the MIC and MBC units of concentration. You state mg/mL (milligrams/mL) in the figure texts and µg/mL in the main text. Regardless, these concentration units should all be converted to molarity for better comparison between compounds and compared to ciprofloxsacin, bleomycin and cloxacillin (Figure 9)., since compounds 6a – 6f are higher molecular weight than 5a-5j.

We presented the values of MIC and MBC in µg / ml to emphasize the small amounts of the analyzed lactones are cytotoxic to bacterial cells in relation to the three antibiotics we used, and that was the only result of it. It is generally accepted to give such values in µg / ml but if necessary we will present these values in mm for better legibility of the drawings.

  • Line 431 – 434: This sentence is not expressed correctly.
  • The sentence has been corrected
  •  
  • Line 437: what are the differences in these strains? How many replicates? How was the dilution? Is there any standard AB as comparison/positive control?
  • The differences between the hooks have been described in the introduction, the number of repetitions has been included in chapter 2.8 in the materials and methods. The dilutions are described in the resulting section. The positive control was compound number 17 from our previous work on coumarins [5].

  • Line 439 – 440 and 451: Please remove mentions of concentration for MBC/MIC, it cannot be expressed as a concentration, it is purely a ratio.
  • we left the concentration values at this stage because we describe separately the values for MIC and MBC and not the relationship to each other
  •  
  • Line 452 - 476: Correct “10-6 dilution”, “mg/ml-1”, “dilution of 10-5” and the like in this paragraph and the rest of the main text.   done
  • Line 474: remove “a 48-well plate was used”.

the sentence has been deleted

  • Table 2: Indicate that these results are based on Tukey’s multiple comparison test, like you say in line 517 – 518. It’s not immediately obvious if this was the Tukey’s test or Fisher’s LSD test.
  • the tukey test was applied
  • Move “covalently closed circle” and “open circle” from lines 496-497 to line 485 where “ccc” and “oc” are first mentioned.

The sentences has been corrected

  • Line 576: remove “she” and change “proves” to something like “shows” or “exhibits” etc.
  • The sentences has been corrected

  • Line 587 – 588: “… antibiotic. smirów (disrupted forms of bacterial DNA in the analyzed material. This shows that…” What is this?
  • The sentences has been corrected
  •  
  • Line 593: Change to “base excision repair system.”

The sentences has been corrected

  • Improve graphs, especially label on axis, but also description
  • we used raw data for the graphs, please indicate to us what exactly the corrections should consist of?

Discussion

  • Lines 604 – 608 are expressed poorly. Improve writing.
  • The sentences has been corrected
  • Line 611: “Highest lues in both types of MIC and MBC tests”? Don’t you mean lowest? Also, you need to check if this is true after changing these values to molarity concentration units.
  • We means about “values”
  • Lines 617 – 620: References [5-9] do not support this statement. Make the appropriate citations of the correct articles to support this.
  • The sentences has been corrected
  • Line 652: remove “and strength”.
  • Has been removed
  •  
  • Lines 669 – 670: correction, “aureus”, remove “48-well”.
  • Has been removed
  •  
  • Line 680 - 682: saying “all strains of pathogenic bacteria” is far too overreaching since you only tested it on coli, make a more conservative statement. Remove quotation marks around “antibiotics”.
  • Has been removed

  • Line 696: italicise Sacchromyces cerevisiae.

  • Has been done
  •  
  • Line 708: “He”? 
  • herpin
  • Remove sentence in Line 738 – 739: “Our research shows that both… all analyzed compounds (Fig. 4-6).”
  •  
  • The sentences has been removed
  • Line 747 – 766: I disagree with this argument about Fluorine ions. Having fluoryl groups instead of H atoms can make a huge difference to the activity of the drug and does not necessarily mean there is reactions of lactones producing free fluoride ions. This would need to be shown by fluoride determination assays after lactone treatment of the coli.

We agree with this thesis, at this stage we have also demonstrated it with such research

  • It does not really become clear how this lactones can be used as potential antibiotic treatments. Only one bacteria species was looked at and only in a very basic way without any toxicity studies in (human) cells.
  • We did not do research on human cells because we considered, as in our previous studies on other cited compounds, one species of E.coli bacteria with basic model strains with and without LPS for which we do not need the approval of the bioethics committee is enough.

Conclusions

  • Conclusions need to be more succinct and directly relevant to the results in this paper.
  • Has been corrected

Reviewer 2 Report

While an interesting paper, there are quite a few issues that need to be addressed:

  1. There are significant spelling/language issues. For instance, a few times lactone is misspelled as lacton. Also I could not understand the paragraph between lines 426 – 440; it needs to be rewritten.
  2. Figure 2 could be prepared better, it is confusing with 5 and 6 just next to each other as if they both came out of the same reaction.
  3. In the text of 3.2 it says the MIC values are in microgram/mL but in the accompanying figures it says mg/mL – I’m not sure which it is but these need to be consistent. If it is mg/mL I am concerned these compounds are not significant.
  4. Similarly to the point above, why is the MIC in figure 9 in mM? The MICs need to be consistent.
  5. I am struggling to understand the point of the second half of section 3.3, do you want to give these lactones synergistically with cloxacillin?
  6. Did you do any studies on if the lactones are toxic to normal cells? It might be helpful to see.
  7. The conclusion section does not introduce the other antibiotics at all, this was a large portion of the paper so it should be in the conclusions.
  8. Why was the only SAR change that was made halogenation? Would like to see this reasoning addressed.

Author Response

Reviewer 2

Thank you very much for the substantive suggestions that will contribute to increasing the quality and scientific value of our manuscript.

While an interesting paper, there are quite a few issues that need to be addressed:

  1. There are significant spelling/language issues. For instance, a few times lactone is misspelled as lacton. Also I could not understand the paragraph between lines 426 – 440; it needs to be rewritten.

We are very grateful for this suggestion. Misspelling of lactone was corrected within manuscript body.

The respective paragraph was corrected. 

  1. Figure 2 could be prepared better, it is confusing with 5 and 6 just next to each other as if they both came out of the same reaction.

We are very grateful for this suggestion. Figure 2 was modified to make its contents clear to readers.

  1. In the text of 3.2 it says the MIC values are in microgram/mL but in the accompanying figures it says mg/mL – I’m not sure which it is but these need to be consistent. If it is mg/mL I am concerned these compounds are not significant.

Has been corrected

  1. Similarly to the point above, why is the MIC in figure 9 in mM? The MICs need to be consistent.

Has been corrected

  1. I am struggling to understand the point of the second half of section 3.3, do you want to give these lactones synergistically with cloxacillin?

It is a complementary study to the previously used MIC and MBC methods. In a study using the Fpg enzyme, which is a bifunctional glycosylase of the Base excison repair system, we modified bacterial DNA with both the analyzed compounds and the antibiotics we analyzed in order to show the degree and intensity of damage to the genetic material by lactones more than the antibiotics used.

  1. Did you do any studies on if the lactones are toxic to normal cells? It might be helpful

Normal cell studies were carried out using the K12 strain itself, which is a typical LPS-free non-pathogenic E.coli strain.to see

  1. The conclusion section does not introduce the other antibiotics at all, this was a large portion of the paper so it should be in the conclusions.

this has been corrected

  1. Why was the only SAR change that was made halogenation? Would like to see this reasoning addressed.

Similar lactone without halogen substituent were already investigated.  We will pay more attention to this in our next research. Thank you very much for the apt suggestion on this issue.

Reviewer 3 Report

In this manuscript authors have synthesized and studied the biological activities of 6 membered lactones. The study is interesting but not very novel. Manuscript could have been shaped in a better way. Following are my input about this manuscript.

  • There are lots of irrelevant texts in the introduction section which needs improvement. Author should mention the MIC and MBC in micromolar concentration instead of ug/ml.
  • Did you test these compounds on gram positive strains as well? What was the main purpose of using gram negative strains only?
  • Why didn’t you include any resistant strains?
  • What is the toxicity on human cells?
  • There are lots of grammatical error and typos throughout the manuscript which needs improvement.
  • NMR and MS spectra are missing in supporting data.
  • NMR characterization should start from aromatic region instead of aliphatic one.
  • What was the purpose of using only halogen as substituents in their molecules from the biological activity point of view?
  • What was the purpose of choosing bleomycin as reference as it is not solely used for bacterial infections?

Author Response

Reviewer 3

Thank you very much for the substantive suggestions that will contribute to increasing the quality and scientific value of our manuscript.

In this manuscript authors have synthesized and studied the biological activities of 6 membered lactones. The study is interesting but not very novel. Manuscript could have been shaped in a better way. Following are my input about this manuscript.

  • There are lots of irrelevant texts in the introduction section which needs improvement. Author should mention the MIC and MBC in micromolar concentration instead of ug/ml.

The introduction has been thoroughly shortened and corrected by us. We presented the values of MIC and MBC in µg / ml to emphasize the small amounts of the analyzed lactones are cytotoxic to bacterial cells in relation to the three antibiotics we used, and that was the only result of it. It is generally accepted to give such values in µg / ml but if necessary we will present these values in mm for better legibility of the drawings.

  • Did you test these compounds on gram positive strains as well? What was the main purpose of using gram negative strains only?

We did not use them on gram-positive strains because these bacteria in their structure do not have an outer cell membrane and the cell wall of gram-positive bacteria is thicker than that of gram-negative bacteria. And we first wanted to check the toxicity of the analyzed lactones on gram-negative strains in order to investigate the mechanism of damaging bacterial membranes. Is it the same or different for the various relationships we analyzed earlier (quoted references 5 to 9 in the Literature chapter of our manuscript). Knowing the change in the value of the oxidation-reduction potential of membranes and the degree of its damage to individual components that build it, along with the LPS contained in it, we can determine whether the individual compounds analyzed by us show a toxic effect on model bacterial cells or not. In order to be able to develop better, more potent drugs in the future, with a stronger effect on resistant bacteria than the currently used antibiotics. Only on the basis of the results of our research, we will be able to use similar methods on gram-positive strains that are less virulent than gram-negative bacteria and compare the mechanisms of action of the analyzed compounds.

  • Why didn’t you include any resistant strains?
  • In our study, we used the K12 model strain which does not have LPS and is a traditional pathogen causing nosocomial infections leading to sepsis. K12 is a resistant strain as presented in the publication by Maciejewska et al., Reference No. 11 in the literature.
  • What is the toxicity on human cells?

The selection of human cells and the testing of lactones on them is in our next project that we want to present soon. Due to the large amount of data and the focus on the mechanisms of lactone toxicity on model bacteria, we have not included the data from these studies here, as they will be part of a larger whole in another study.

  • There are lots of grammatical error and typos throughout the manuscript which needs improvement.
  • we have tried to correct all grammatical inaccuracies
  • NMR and MS spectra are missing in supporting data.
  • All spectra are incorporated in manuscript body according to Journal guidance to authors.

  • NMR characterization should start from aromatic region instead of aliphatic one.
  • This is not obligatory according to Journal guidance to authors.
  • What was the purpose of using only halogen as substituents in their molecules from the biological activity point of view?

We wanted to see how individual types of substituents in the lactone structure as a whole will act on model bacterial strains. Are there any preferences for specific substituents and their biological / cytotoxic activity on the analyzed bacterial cells?

  • What was the purpose of choosing bleomycin as reference as it is not solely used for bacterial infections?

In our research, we used one of the three antibiotics - bleomycin, which is a glycopeptide antibiotic obtained from the Streptomyces verticillus strain. It is a mixture of polypeptide compounds with cytostatic activity, mainly bleomycin A₂ and B₂. The mechanism of its action is based on the binding of bleomycin molecules in a complex with iron (II) ions to single and double DNA strands at the site of precisely defined nucleotide sequences and causing single and double breaks of these strands in the presence of oxygen, which in turn inhibits the synthesis of genetic material, division and cell growth, eventually leading to the initiation of apoptotic processes or aging of tumor cells. Cells in the G2 and M phase of the cell cycle are the most sensitive. The potency of bleomycin varies across tissues and cells, including due to the variation in the content of hydrolases that break down the molecules of this antibiotic in cell proliferative processes. It is known from the literature that tumors with a greater degree of differentiation usually react more strongly to the action of bleomycin than anaplastic tumors. Squamous cells, in which the degree of bleomycin hydrolysis is mostly low, show a high sensitivity to bleomycin. In tissues more sensitive to the action of this antibiotic and in normal tumor tissues, bleomycin often results in chromosomal abnormalities such as fragmentation, chromatid disruption and translocation. In our research we used bleomycin knowing that it can totally damage bacterial DNA, so we treated it as a positive control for lactones. It turned out that the modification of bleomycin itself, like other antibiotics, is not strong enough to damage the bacterial DNA to a large extent. A greater effect was visible after the use of the analyzed lactones.

Round 2

Reviewer 2 Report

Thank you for making the suggested changes, the paper is better. There are still a few suggestions:

  1. The language and spelling are better but still need more work. I suggest you check it again as some parts are still unreadable.
  2. Thank you for addressing my point about normal cells, however I meant eukaryotic cells. If these are toxic to humans, then there is really no point in investigating them as potential antibiotics. K12 does not address this issue.
  3. I am not sure of the point of lines 770 – 781 – yes fluoride ions are toxic but these compounds do not have them. The fluorine in the structure will not be released so I’m not sure as to the point of this. However the electronegativity is relevant.
  4. As you define the ring size of gamma and delta lactones in the introduction, it might be nice to define the ring size of alpha and beta as well.
  5. Why did you only introduce cloxacillin but not ciprofloxacin or bleomycin?

Author Response

Thank you very much for the substantive suggestions and very valuable comments that contributed to the improvement of the quality and scientific value of our manuscript.

Thank you for making the suggested changes, the paper is better. There are still a few suggestions:

  1. The language and spelling are better but still need more work. I suggest you check it again as some parts are still unreadable.

We tried to improve the language as much as we could to make it more understandable for the readers.

  1. Thank you for addressing my point about normal cells, however I meant eukaryotic cells. If these are toxic to humans, then there is really no point in investigating them as potential antibiotics. K12 does not address this issue. 

Thank you very much for this suggestion, but we have already included it in our future research plans. Currently, we have not conducted studies on eukaryotic cells using the normal model strain K12 with its mutants R2-R4 to demonstrate the antibacterial activity of the analyzed compounds. So far, research on this type of compounds seems to be very promising for the use of these compounds as less toxic analogues of many commonly used antibiotics. Our next project is to conduct tests on eukaryotic cells to determine the presence or absence of toxic effects of the analyzed compounds. At present, there are no literature data on this subject, therefore there is a need to clarify their role.

  1. I am not sure of the point of lines 770 – 781 – yes fluoride ions are toxic but these compounds do not have them. The fluorine in the structure will not be released so I’m not sure as to the point of this. However the electronegativity is relevant.

An extensive paragraph was added in the discussion on the toxicity of fluoride as an element on eukaryotic cells, marked with azure color

  1. As you define the ring size of gamma and delta lactones in the introduction, it might be nice to define the ring size of alpha and beta as well.

Thank you for your valuable comment. The ring size of alpha and beta lactones is now define on page 2. 

  1. Why did you only introduce cloxacillin but not ciprofloxacin or bleomycin?

We took into account the short bactericidal activity of both antibiotics, we apologize for our oversight.

              The corrections were included in the manuscript and marked in azure in the text.

Reviewer 3 Report

After going through the revised manuscript and author's response, I couldn't see significant improvement in the manuscript. I would recommend publishing in its current form. 

Author Response

After going through the revised manuscript and author's response, I couldn't see significant improvement in the manuscript. I would recommend publishing in its current form. 

Thank you very much for the substantive suggestions and very valuable comments that contributed to the improvement of the quality and scientific value of our manuscript.

We added an extensive paragraph about the role of fluoride on eukaryotic cells and we briefly characterized the antibiotics that were used by us.  All corrections have been marked in azure by us.
